# Taylor TD-learning

**Michele Garibbo**
Department of Engineering Mathematics
University of Bristol
Bristol, United Kingdom
michele.garibbo@bristol.ac.uk

**Maxime Robeyns**
Department of Engineering Mathematics
University of Bristol
Bristol, United Kingdom
maxime.robeyns.2018@bristol.ac.uk

**Laurence Aitchison**
Department of Engineering Mathematics
University of Bristol
Bristol, United Kingdom
laurence.aitchison@bristol.ac.uk

## Abstract

Many reinforcement learning approaches rely on temporal-difference (TD) learning to learn a critic. However, TD-learning updates can be high variance. Here, we introduce a model-based RL framework, Taylor TD, which reduces this variance in continuous state-action settings. Taylor TD uses a first-order Taylor series expansion of TD updates. This expansion allows Taylor TD to analytically integrate over stochasticity in the action-choice, and some stochasticity in the state distribution for the initial state and action of each TD update. We include theoretical and empirical evidence that Taylor TD updates are indeed lower variance than standard TD updates. Additionally, we show Taylor TD has the same stable learning guarantees as standard TD-learning with linear function approximation under a reasonable assumption. Next, we combine Taylor TD with the TD3 algorithm, forming TaTD3. We show TaTD3 performs as well, if not better, than several state-of-the art model-free and model-based baseline algorithms on a set of standard benchmark tasks.

## 1 Introduction

Actor-critic algorithms underlie many of the recent successes of deep RL [1, 2, 3, 4, 5, 6, 7, 8]. In these algorithms, the actor provides the control policy while the critic estimates the policy's expected long-term returns [i.e. the value function; 9, 10]. The critic is typically trained using some form of temporal-difference (TD) update [e.g. 3, 7, 1, 2, 6]. These TD updates need to be computed in expectation over a large distribution of visited states and actions, induced by the policy and the environment dynamics [11, 12]. Since this expectation is analytically intractable, TD updates are typically performed based on individually sampled state-action pairs from real environmental transitions (i.e. sample-based estimates). However, the variance of (sample-based) TD updates can be quite large, meaning that we need to average over many TD updates for different initial states and actions to get a good estimate of the expected updates [13].

Model-based strategies provide a promising candidate to tackle this high variance [14]. For instance, Dyna methods, among the most popular model-based strategies, use a learned model of the environment transitions to generate additional imaginary transitions. These imaginary transitions can be used as extra training samples for TD methods [e.g. 15, 16, 17, 18, 19, 20]. Although the additional (imaginary) transitions help in reducing the variance of performing TD-updates over a

large distribution of state-action pairs, Dyna methods still rely on the same, potentially high-variance (sample-based) TD-updates as standard TD-learning.

We address the issue of high-variance TD-updates by formulating an expected TD-update over a small distribution of state-action pairs. We show this expected update can analytically be estimated with a first-order Taylor expansion for continuous state-action spaces, in an approach we call *Taylor TD*. By analytically estimating this expected update, rather than exclusively relying on sample-based estimates (as in e.g. Dyna), we get lower variance TD updates. Additionally, we show Taylor TD does not affect the stable learning guarantees of TD-learning under linear function approximation [for a fixed policy as shown by 21]. Next, we propose a model-based off-policy algorithm, Taylor TD3 (TaTD3), which uses Taylor TD in combination with the TD3 algorithm [1]. We show TaTD3 performs as well as if not better than several state-of-the art model-free and model-based baseline algorithms on a set of standard benchmark tasks. Finally, we compare TaTD3 to its "Dyna" equivalent, which exclusively relies on sample-based TD-updates. We found the largest benefits of Taylor TD may appear in high dimensional state-action spaces.

## 2 Background

Reinforcement learning aims to learn reward-maximising behaviour by interacting with the surrounding environment. At each discrete time step, $t$, the agent in state $\mathbf{s} \in \mathcal{S}$, chooses an action $\mathbf{a} \in \mathcal{A}$ based on a policy $\pi : \mathcal{S} \to \mathcal{A}$, and observes a scalar reward, $r$ and a new state $\mathbf{s}' \in \mathcal{S}$ from the environment. The agent's goal is to find the policy that maximises the expected sum of rewards (i.e. the expected return), from a distribution of initial states (or state-action pairs). As such, it is usually necessary to compute the expected return for a state-action pair $(\mathbf{s}, \mathbf{a})$ and a policy $\pi$; which we can do with a value function. Given a policy $\pi$ and an initial state-action pair $(\mathbf{s}, \mathbf{a})$, we define the value function $Q^\pi(\mathbf{s}, \mathbf{a}) = \mathrm{E}\left[R_t \mid S_t = \mathbf{s}, A_t = \mathbf{a}\right]$, where $R_t = \sum_{i=t}^{T} \gamma^{i-t} r_i$ is the discounted sum of future rewards from the current time step $t$ until termination $T$, with discount factor $\gamma \in [0, 1]$. The value function or critic, $Q^\pi$, quantifies how good the policy, $\pi$, is in terms of its expected return when taking action $\mathbf{a}$ in state $\mathbf{s}$ and following the policy $\pi$ thereafter.

In off-policy learning, a function approximation, $Q_\theta$, is typically used to approximate the true value function, $Q^{\pi^{\text{target}}}$ of a target policy, $\pi^{\text{target}}$, based on experience collected from behavioural policies, $\pi^{\text{behave}}$. The function approximation is typically updated with some form of TD-learning [11, 22],

$$\mathrm{E}_{(s,a)\sim\mathcal{B}}\left[\Delta\theta(\mathbf{s}, \mathbf{a})\right] = \mathrm{E}_{(s,a)\sim\mathcal{B}}\left[\delta_\theta(\mathbf{s}, \mathbf{a})\nabla_\theta Q_\theta(\mathbf{s}, \mathbf{a})\right], \tag{1}$$

$$\delta_\theta(\mathbf{s}, \mathbf{a}) = r(\mathbf{s}, \mathbf{a}) + \gamma Q_\theta(\mathbf{s}', \mathbf{a}') - Q_\theta(\mathbf{s}, \mathbf{a}). \tag{2}$$

where $\mathcal{B}$ denotes a reply buffer containing a mixture of states and actions sampled according to behavioural policies, $\pi^{\text{behave}}$. Note that we have not included a learning rate in these equations, so $\Delta\theta(\mathbf{s}, \mathbf{a})$ only represents an update direction, and not the update magnitude. Additionally, in the tuple $(\mathbf{s}, \mathbf{a}, \mathbf{s}', \mathbf{a}')$, we are gonna refer to $\mathbf{s}$ as the initial state, $\mathbf{a}$ as the initial action, $\mathbf{s}'$ as the next state and $\mathbf{a}'$ as the target action. In standard (model-free) off-policy learning, the initial state and action, as well as the reward and the next state $(\mathbf{s}, \mathbf{a}, r, \mathbf{s}')$ are drawn from the replay buffer [e.g., 23], while the next action comes from the target policy, $\mathbf{a}' \sim \pi^{\text{target}}(\mathbf{s}')$. Critically, this off-policy setting allows to learn $Q^{\pi^{\text{target}}}$, even when the initial states and actions, $(\mathbf{s}, \mathbf{a})$ from the replay buffer, come from a different policy (or even an unknown policy).

In the model-based setting, we can exploit this flexibility in the distribution over the initial state, $\mathbf{s}$, and action, $\mathbf{a}$. Specifically, we sample the initial state, $\mathbf{s}$ from an initial-state distribution, $d^{\text{init}}$ (if we are taking initial states from the replay buffer, then $d^{\text{init}}$, would be the distribution over states in the buffer). Next, we sample the initial action, $\mathbf{a}$, from an arbitrary initial policy, $\pi^{\text{init}}(\mathbf{s})$ (i.e., the initial action, $\mathbf{a}$, does not have to come from the buffer as for the model-free case). Then, we sample the reward and next state $(r, \mathbf{s}')$ from the (learned) model, and finally we sample the next action from the target policy,

$$\mathbf{a} \sim \pi^{\text{init}}(\mathbf{s}) \qquad r, \mathbf{s}' \sim \text{learned-model}(\mathbf{s}, \mathbf{a}) \qquad \mathbf{a}' \sim \pi^{\text{target}}(\mathbf{s}'). \tag{3}$$

In principle, the off-policy setting allows for complete freedom in the distribution over the initial state, $d^{\text{init}}$, and the initial policy, $\pi^{\text{init}}$. That freedom is necessary in the model-free off-policy setting, because the states and actions in the replay buffer often come from an unknown distribution (e.g., all past behavioural policies). Although most model-based approaches take the initial policy $\pi^{\text{init}}$ to

equal either the target $\pi^{\text{target}}$ or behavioural policy $\pi^{\text{behave}}$ (i.e., the policy used to interact with the environment) [e.g., 19, 18], this does not have to be the case. For instance, $\pi^{\text{init}}$ could be a broader distribution than $\pi^{\text{behave}}$, enabling us to update and improve the action value function for a broader range of actions.

## 3 Taylor TD

The expected update over actions in Eq. (1) is usually intractable, due to the complexity of the function, $\delta_\theta(\mathbf{s}, \mathbf{a})\nabla_\theta Q_\theta(\mathbf{s}, \mathbf{a})$. Standard TD-learning methods therefore employ a sample-based approach, by sampling initial actions from some policy (distribution) $\pi^{\text{init}}$ and performing the (TD) updates for those actions. However, sampling can give rise to high-variance estimators, and hence slow learning [24]. Taylor TD reduces this variance by introducing an analytic approximation to the expectation in Eq. (1) over some stochasticity in the distribution over actions (and over the state, i.e., Sec. 3.2). This approach requires a differentiable model of the transitions and rewards as well as continuous state and action space.

Taylor TD takes advantage of the flexibility in choice of the initial, target and behavioural policies arising in model-based off-policy settings (see Sec. 2). Taylor TD can use any target and behavioural policy, but assumes some structure in the initial policy, $\pi^{\text{init}}$. Specifically, Taylor TD splits the process of sampling the initial action into two steps. It first samples the mean, $\boldsymbol{\mu}_{\text{a}}$, from a mean-policy, $\pi^{\text{action-mean}}(\mathbf{s})$ (usually, we would use the target policy, $\pi^{\text{target}}$, as the mean-policy, but we do not have to). Then Taylor TD samples "noise" from a second distribution, $\pi^{\text{action-noise}}$, and the actual (initial) action is the sum of the mean and the noise,

$$\boldsymbol{\mu}_{\text{a}} \sim \pi^{\text{action-mean}}(\mathbf{s}) \qquad\qquad \boldsymbol{\xi}_{\text{a}} \sim \pi^{\text{action-noise}} \qquad (4)$$

$$\mathbf{a} = \boldsymbol{\mu}_{\text{a}} + \boldsymbol{\xi}_{\text{a}}. \qquad (5)$$

Here, we require that the noise, $\boldsymbol{\xi}_{\text{a}}$, drawn from $\pi^{\text{action-noise}}$ has zero expectation, and finite, known covariance, $\boldsymbol{\Sigma}_{\text{a}}$,

$$\mathrm{E}_{\boldsymbol{\xi}_{\text{a}}}[\boldsymbol{\xi}_{\text{a}}] = \mathbf{0} \qquad\qquad \mathrm{E}_{\boldsymbol{\xi}_{\text{a}}}\left[\boldsymbol{\xi}_{\text{a}}\boldsymbol{\xi}_{\text{a}}^T\right] = \boldsymbol{\Sigma}_{\text{a}}. \qquad (6)$$

The overall initial policy, combining these two steps, can be written, $\mathbf{a} \sim \pi^{\text{init}}(\mathbf{s})$.

### 3.1 Action expansion

Here, we analytically approximate the expectation over the action-noise, $\boldsymbol{\xi}_{\text{a}}$, using a first-order Taylor expansion. Specifically, we define the expected TD update, averaging over the action noise, $\boldsymbol{\xi}_{\text{a}}$, as,

$$\Delta_{\text{Exp}}\theta(\mathbf{s}, \boldsymbol{\mu}_{\text{a}}) \coloneqq \mathrm{E}_{\boldsymbol{\xi}_{\text{a}}}\left[\Delta\theta(\mathbf{s}, \boldsymbol{\mu}_{\text{a}} + \boldsymbol{\xi}_{\text{a}})|\mathbf{s}, \boldsymbol{\mu}_{\text{a}}\right] = \mathrm{E}_{\boldsymbol{\xi}_{\text{a}}}\left[\delta_\theta(\mathbf{s}, \boldsymbol{\mu}_{\text{a}} + \boldsymbol{\xi}_{\text{a}})\nabla_\theta Q(\mathbf{s}, \boldsymbol{\mu}_{\text{a}} + \boldsymbol{\xi}_{\text{a}})\Big|\mathbf{s}, \boldsymbol{\mu}_{\text{a}}\right] \quad (7)$$

The (overall) expected TD update in Eq. 1 can now be written in terms of $\Delta_{\text{Exp}}\theta(\mathbf{s}, \boldsymbol{\mu}_{\text{a}})$,

$$\mathrm{E}_{\mathbf{s}, \mathbf{a}}\left[\Delta\theta(\mathbf{s}, \mathbf{a})\right] = \mathrm{E}_{\mathbf{s}, \boldsymbol{\mu}_{\text{a}}, \boldsymbol{\xi}_{\text{a}}}\left[\Delta\theta(\mathbf{s}, \boldsymbol{\mu}_{\text{a}} + \boldsymbol{\xi}_{\text{a}})\right] = \mathrm{E}_{\mathbf{s}, \boldsymbol{\mu}_{\text{a}}}\left[\Delta_{\text{Exp}}\theta(\mathbf{s}, \boldsymbol{\mu}_{\text{a}})\right] \qquad (8)$$

The exact expectation for $\Delta_{\text{Exp}}\theta(\mathbf{s}, \boldsymbol{\mu}_{\text{a}})$ (Eq. 7) is also not tractable typically, due to the complexity of the function, $\delta_\theta(\mathbf{s}, \boldsymbol{\mu}_{\text{a}} + \boldsymbol{\xi}_{\text{a}})\nabla_\theta Q(\mathbf{s}, \boldsymbol{\mu}_{\text{a}} + \boldsymbol{\xi}_{\text{a}})$. Instead, Taylor TD uses $\Delta_{\text{Ta}}\theta(\mathbf{s}, \boldsymbol{\mu}_{\text{a}})$ to approximate $\Delta_{\text{Exp}}\theta(\mathbf{s}, \boldsymbol{\mu}_{\text{a}})$ (Eq. 7),

$$\Delta_{\text{Ta}}\theta(\mathbf{s}, \boldsymbol{\mu}_{\text{a}}) \approx \Delta_{\text{Exp}}\theta(\mathbf{s}, \boldsymbol{\mu}_{\text{a}}) \qquad (9)$$

Specifically, $\Delta_{\text{Ta}}\theta(\mathbf{s}, \boldsymbol{\mu}_{\text{a}})$, arises from a first-order Taylor expansion around $\boldsymbol{\xi}_{\text{a}} = \mathbf{0}$,

$$\Delta_{\text{Ta}}(\mathbf{s}, \boldsymbol{\mu}_{\text{a}}) = \mathrm{E}_{\boldsymbol{\xi}_{\text{a}}}\left[\left(\delta_\theta(\mathbf{s}, \boldsymbol{\mu}_{\text{a}}) + \boldsymbol{\xi}_{\text{a}}^T\nabla_{\mathbf{a}}\delta_\theta(\mathbf{s}, \boldsymbol{\mu}_{\text{a}})\right)\nabla_\theta\left(Q_\theta(\mathbf{s}, \boldsymbol{\mu}_{\text{a}}) + \boldsymbol{\xi}_{\text{a}}^T\nabla_{\mathbf{a}}Q_\theta(\mathbf{s}, \boldsymbol{\mu}_{\text{a}})\right)\Big|\mathbf{s}, \boldsymbol{\mu}_{\text{a}}\right] \qquad (10)$$

This reduces to (see Appendix A.1),

$$\Delta_{\text{Ta}}\theta(\mathbf{s}, \boldsymbol{\mu}_{\text{a}}) = \delta_\theta(\mathbf{s}, \boldsymbol{\mu}_{\text{a}})\nabla_\theta Q_\theta(\mathbf{s}, \boldsymbol{\mu}_{\text{a}}) + \nabla_{\theta, \mathbf{a}}^2 Q_\theta(\mathbf{s}, \boldsymbol{\mu}_{\text{a}})\boldsymbol{\Sigma}_{\text{a}}\nabla_{\mathbf{a}}\delta_\theta(\mathbf{s}, \boldsymbol{\mu}_{\text{a}}). \qquad (11)$$

For simplicity, we further assume that the covariance of initial actions is isotropic, $\mathbf{\Sigma}_{\mathrm{a}} = \lambda_{\mathrm{a}}\mathbf{I}$ (although this is not a requirement), resulting in the following (1st-order) Taylor TD-update:

$$\Delta_{\mathrm{Ta}}\theta(\mathbf{s}, \boldsymbol{\mu}_{\mathrm{a}}) = \delta_{\theta}(\mathbf{s}, \boldsymbol{\mu}_{\mathrm{a}})\nabla_{\theta}Q_{\theta}(\mathbf{s}, \boldsymbol{\mu}_{\mathrm{a}}) + \lambda_{\mathrm{a}}\nabla^2_{\theta,\mathbf{a}}Q_{\theta}(\mathbf{s}, \boldsymbol{\mu}_{\mathrm{a}})\nabla_{\mathbf{a}}\delta_{\theta}(\mathbf{s}, \boldsymbol{\mu}_{\mathrm{a}}) \qquad (12)$$

This Taylor TD-update includes the standard TD update at state $\mathbf{s}$ and (mean) action $\boldsymbol{\mu}_{\mathrm{a}}$ plus a new term, which tries to align the action gradient of the critic (Q-function) with the action gradient of the TD target. Conceptually, this gradient matching should help reduce the variance across TD-updates since it provides a way to analytically integrate over some of the stochasticity induced by $\pi^{\mathrm{init}}$. When doing a first order Taylor series approximation, we may worry that errors in the Taylor expansion might affect convergence. In the Appendix B, we include a proof that for linear function approximation, any errors in the first-order Taylor expansion do not affect the stability of TD-learning (assuming small simulator time steps). Importantly, this result is not a trivial, as there are still approximation errors in the mapping from state and action to the features that we use for linear function approximation. Moreover, we provide empirical evidence that the first-order Taylor expansion reduces the variance of standard TD-updates and support efficient learning, even under non-linear function approximation (see Sections, 3.4, and 4.2.3).

## 3.2 State expansion

We are not limited to analytically approximate the expectation in Eq. (1) for the distribution of initial actions, but we can extend this approach to a distribution of initial states. Namely, instead of performing a TD-update at the single initial state, $\mathbf{s}$, we perform this update over a distribution of states. Similarly to the action expansion, we assume structure in the distribution over the initial state distribution (denoted here as $d^{\mathrm{init}}$). Specifically, sampling initial states from the initial state distribution involves two steps. We first sample the mean of $d^{\mathrm{init}}$, $\boldsymbol{\mu}_{\mathrm{s}}$, from a state-mean distribution $d^{\mathrm{state\text{-}mean}}$ (usually, we would use the distribution of states in a reply buffer as the state-mean distribution, but we do not have to). Then we sample state-noise from a second distribution, $\boldsymbol{\xi}^{\mathrm{state\text{-}noise}}$, and the actual (initial) state is the sum of the state-mean and the state-noise,

$$\boldsymbol{\mu}_{\mathrm{s}} \sim d^{\mathrm{state\text{-}mean}} \qquad\qquad \boldsymbol{\xi}_{\mathrm{s}} \sim d^{\mathrm{state\text{-}noise}} \qquad (13)$$

$$\mathbf{s} = \boldsymbol{\mu}_{\mathrm{s}} + \boldsymbol{\xi}_{\mathrm{s}}. \qquad (14)$$

Again, we require that the state-noise, $\boldsymbol{\xi}_{\mathrm{s}}$, drawn from $d^{\mathrm{state\text{-}noise}}$ has zero expectation, and finite, known covariance,

$$\mathrm{E}_{\boldsymbol{\xi}_{\mathrm{s}}}[\boldsymbol{\xi}_{\mathrm{s}}] = \mathbf{0} \qquad\qquad \mathrm{E}_{\boldsymbol{\xi}_{\mathrm{s}}}\left[\boldsymbol{\xi}_{\mathrm{s}}\boldsymbol{\xi}_{\mathrm{s}}^T\right] = \mathbf{\Sigma}_{\mathrm{s}}. \qquad (15)$$

Thus, we can again formulate an expected TD-update, averaging over the state-noise,

$$\mathrm{E}_{\boldsymbol{\xi}_{\mathrm{s}}}[\Delta\theta(\boldsymbol{\mu}_{\mathrm{s}} + \boldsymbol{\xi}_{\mathrm{s}}, \mathbf{a})|\boldsymbol{\mu}_{\mathrm{s}}, \mathbf{a}] = \mathrm{E}_{\boldsymbol{\xi}_{\mathrm{s}}}[\delta_{\theta}(\boldsymbol{\mu}_{\mathrm{s}} + \boldsymbol{\xi}_{\mathrm{s}}, \mathbf{a})\nabla_{\theta}Q_{\theta}(\boldsymbol{\mu}_{\mathrm{s}} + \boldsymbol{\xi}_{\mathrm{s}}, \mathbf{a})|\boldsymbol{\mu}_{\mathrm{s}}, \mathbf{a}] \qquad (16)$$

Again, we can approximate this expected update with a first-order Taylor approximation, but this time, we expand the states around $\boldsymbol{\xi}_{\mathrm{s}} = 0$ instead of the actions. This requires that the states are continuous, which is usually the case for control tasks. Based on a similar derivation to the action expansion, we get the following Taylor TD-update, where we assumed the state covariance over TD updates to be isotropic, $\mathbf{\Sigma}_{\mathrm{s}} = \lambda_{\mathrm{s}}\mathbf{I}$ (although again this is not a requirement) (see Appendix A.2 for the full derivation):

$$\Delta_{\mathrm{Ta}}\theta(\boldsymbol{\mu}_{\mathrm{s}}, \mathbf{a}) = \delta_{\theta}(\boldsymbol{\mu}_{\mathrm{s}}, \mathbf{a})\nabla_{\theta}Q_{\theta}(\boldsymbol{\mu}_{\mathrm{s}}, \mathbf{a}) + \lambda_{s}\nabla^2_{\theta,\mathbf{s}}Q_{\theta}(\boldsymbol{\mu}_{\mathrm{s}}, \mathbf{a})\nabla_{\mathbf{s}}\delta_{\theta}(\boldsymbol{\mu}_{\mathrm{s}}, \mathbf{a}) \qquad (17)$$

The rational behind this update is trying to tackle some of the TD-update variance induced by the initial state distribution (e.g., the distribution of sates in a reply buffer), although we expect this only to work for states close-by to the visited ones (i.e. for small values of $\lambda_{\mathrm{s}}$).

## 3.3 State-Action expansion implementation

Finally, we can combine the two Taylor expansions into a single TD-update involving both state and action expansions. Nevertheless, computing the dot products between $\nabla\delta_{\theta}$ and $\nabla Q_{\theta}$ terms for both state and action terms may not be optimal. One reason for this is dot products are unbounded, increasing the risk of high variance (TD) updates [e.g. 25]. To tackle this issue, we use cosine distances between the gradient terms instead of dot products (see Appendix K.2 for the benefits of

**Algorithm 1:** Taylor TD

---

Initialise reply buffer $\mathcal{B}$
Initialise model, critic and target policy parameters $\{w, \theta, \phi\}$
Initialise $\zeta_{\mathrm{a}}, \zeta_{\mathrm{s}} = 0$
**for** each training step **do**
    Collect transition $(\mathbf{s}, \mathbf{a}, r, \mathbf{s}')$ according to $\pi^{\mathrm{behave}}$,      $\{$e.g., $\pi^{\mathrm{behave}} = \mathcal{N}(\pi_\phi^{\mathrm{det\text{-}target}}, \sigma^2 \mathbf{I})\}$
    $\mathcal{B} \leftarrow \mathcal{B} \cup \mathcal{B}(\mathbf{s}, \mathbf{a}, r, \mathbf{s}')$
    **for** each model update step **do**
        $w \leftarrow w - \eta_m \nabla_w \mathcal{L}_w^{\mathrm{model}}(\mathbf{s}, \mathbf{a}, r, \mathbf{s}')$,      $(\mathbf{s}, \mathbf{a}, r, \mathbf{s}') \sim \mathcal{B}$
    **end for**
    **for** each policy update step **do**
        $(\boldsymbol{\mu}_{\mathrm{s}}, \cdot, \cdot, \cdot) \sim \mathcal{B}$
        $\boldsymbol{\mu}_{\mathrm{a}} = \pi_\phi^{\mathrm{det\text{-}target}}(\boldsymbol{\mu}_{\mathrm{s}})$
        $\hat{r}, \hat{\mathbf{s}}' \sim p_w(\cdot \mid \boldsymbol{\mu}_{\mathrm{s}}, \boldsymbol{\mu}_{\mathrm{a}})$
        $\delta = \hat{r} + \gamma Q_\theta(\hat{\mathbf{s}}', \pi_\phi^{\mathrm{det\text{-}target}}(\hat{\mathbf{s}}')) - Q_\theta(\boldsymbol{\mu}_{\mathrm{s}}, \boldsymbol{\mu}_{\mathrm{a}})$
        **if** Action expansion **then**
            $\zeta_{\mathrm{a}}^{\mathrm{critic}} = \mathrm{CosineSimilarity}(\nabla_{\mathbf{a}} Q_\theta(\boldsymbol{\mu}_{\mathrm{s}}, \boldsymbol{\mu}_{\mathrm{a}}), \ \nabla_{\mathbf{a}} \delta(\boldsymbol{\mu}_{\mathrm{s}}, \boldsymbol{\mu}_{\mathrm{a}}))$
        **end if**
        **if** State expansion **then**
            $\zeta_{\mathrm{s}}^{\mathrm{critic}} = \mathrm{CosineSimilarity}(\nabla_{\mathbf{s}} Q_\theta(\boldsymbol{\mu}_{\mathrm{s}}, \boldsymbol{\mu}_{\mathrm{a}}), \ \nabla_{\mathbf{s}} \delta(\boldsymbol{\mu}_{\mathrm{s}}, \boldsymbol{\mu}_{\mathrm{a}}))$
        **end if**
        $\theta \leftarrow \theta + \eta_c \delta \ \nabla_\theta Q_\theta(\boldsymbol{\mu}_{\mathrm{s}}, \boldsymbol{\mu}_{\mathrm{a}}) + \eta_c \lambda_{\mathrm{a}} \nabla_\theta \zeta_{\mathrm{a}}^{\mathrm{critic}} + \eta_c \lambda_{\mathrm{s}} \nabla_\theta \zeta_{\mathrm{s}}^{\mathrm{critic}}$
        $\phi \leftarrow \phi + \eta_p \nabla_\phi Q_\theta(\boldsymbol{\mu}_{\mathrm{s}}, \boldsymbol{\mu}_{\mathrm{a}})$
    **end for**
**end for**

---

this). The cosine distance has the advantage of being bounded. By putting everything together, we propose a novel TD update, which we express below in terms of a loss:

$$
\begin{aligned}
\mathcal{L}_\theta^{\mathrm{critic}} = & \ \delta(\boldsymbol{\mu}_{\mathrm{s}}, \boldsymbol{\mu}_{\mathrm{a}}) Q_\theta(\boldsymbol{\mu}_{\mathrm{s}}, \boldsymbol{\mu}_{\mathrm{a}}) \\
& + \lambda_{\mathrm{a}} \, \mathrm{CosineSimilarity}(\nabla_{\mathbf{a}} Q_\theta(\boldsymbol{\mu}_{\mathrm{s}}, \boldsymbol{\mu}_{\mathrm{a}}), \ \nabla_{\mathbf{a}} \delta(\boldsymbol{\mu}_{\mathrm{s}}, \boldsymbol{\mu}_{\mathrm{a}})) \\
& + \lambda_{\mathrm{s}} \, \mathrm{CosineSimilarity}(\nabla_{\mathbf{s}} Q_\theta(\boldsymbol{\mu}_{\mathrm{s}}, \boldsymbol{\mu}_{\mathrm{a}}), \ \nabla_{\mathbf{s}} \delta(\boldsymbol{\mu}_{\mathrm{s}}, \boldsymbol{\mu}_{\mathrm{a}}))
\end{aligned}
\tag{18}
$$

Note we used the notation $\delta$ instead of $\delta_\theta$ to indicate we are treating $\delta(\boldsymbol{\mu}_{\mathrm{s}}, \boldsymbol{\mu}_{\mathrm{a}})$ as a fixed variable independent of $\theta$. This ensures when we take the gradient of this loss relative to $\theta$, we do not differentiate through any $\delta$ terms (following the standard implementation of TD-updates in autodiff frameworks such as PyTorch, see Appendix D). In practice, Taylor TD requires a differentiable model of the environment transitions as well as reward function in order to compute the terms $\nabla_{\mathbf{a}} \delta_\theta(\boldsymbol{\mu}_{\mathrm{s}}, \boldsymbol{\mu}_{\mathrm{a}})$ and $\nabla_{\mathbf{s}} \delta_\theta(\boldsymbol{\mu}_{\mathrm{s}}, \boldsymbol{\mu}_{\mathrm{a}})$ (see Appendix C for further details).

In Sections 3.1 and 3.2, we have been careful to setup Taylor TD over actions and states in the most general possible setting. Nevertheless, a set-up that may work well with Taylor TD is following,

$$
\boldsymbol{\mu}_{\mathrm{a}} = \pi^{\mathrm{det\text{-}target}}(\mathbf{s}) \qquad\qquad \boldsymbol{\mu}_{\mathrm{s}} \sim \mathcal{B}
\tag{19}
$$

where values for $\boldsymbol{\mu}_{\mathrm{a}}$ (i.e., the mean-policy) are the outputs of a deterministic (target) policy $\pi^{\mathrm{det\text{-}target}}$ and $\boldsymbol{\mu}_{\mathrm{s}}$ (i.e., the state-means) are set to states sampled from a reply buffer $\mathcal{B}$. In this way, Taylor TD performs TD updates that analytically integrate the expected Q-value of the target policy ($Q^{\pi^{\mathrm{det\text{-}target}}}$) over a distribution of actions centered at $\pi^{\mathrm{det\text{-}target}}$ and controlled by $\lambda_{\mathrm{a}}$ as well as over distributions of states centered at previously visited states (i.e., from the reply buffer) and controlled by $\lambda_{\mathrm{s}}$. We provide a Taylor TD algorithm implementation with this set-up in the Algorithm box 1. Note, in the Algorithm box 1, we left the model loss unspecified (i.e., $\mathcal{L}^{\mathrm{model}}$) to denote that any valid method can be used to learn the model of the transitions. However, in the experiments below, we always learned the transition model based on maximum likelihood on the observed environment transitions (see Appendix E).

### 3.4 Variance analysis

Here, we show that the expected TD update in Eq. (7) (which Taylor TD estimates) has lower variance than standard (sample-based) TD-updates over the same distribution of actions. We only provide this variance analysis for the distribution over actions, because an analogous theorem can be derived for the distribution over states (i.e. Eq. 16). To begin, we apply the law of total variance [26] to standard TD-updates,

$$\text{Var}_{\mathbf{s},\mathbf{a}}[\Delta\theta(\mathbf{s},\mathbf{a})] = \text{Var}_{\mathbf{s},\boldsymbol{\mu}_{\mathrm{a}},\boldsymbol{\xi}_{\mathbf{a}}}[\Delta\theta(\mathbf{s},\boldsymbol{\mu}_{\mathrm{a}}+\boldsymbol{\xi}_{\mathbf{a}})] \tag{20}$$
$$= \text{E}_{\mathbf{s},\boldsymbol{\mu}_{\mathrm{a}}}\left[\text{Var}_{\boldsymbol{\xi}_{\mathrm{a}}}[\Delta\theta(\mathbf{s},\boldsymbol{\mu}_{\mathrm{a}}+\boldsymbol{\xi}_{\mathbf{a}})|\mathbf{s},\boldsymbol{\mu}_{\mathrm{a}}]\right] + \text{Var}_{\mathbf{s},\boldsymbol{\mu}_{\mathrm{a}}}\left[\text{E}_{\boldsymbol{\xi}_{\mathrm{a}}}[\Delta\theta(\mathbf{s},\boldsymbol{\mu}_{\mathrm{a}}+\boldsymbol{\xi}_{\mathbf{a}})|\mathbf{s},\boldsymbol{\mu}_{\mathrm{a}}]\right]$$

The inner expectation and inner variance samples action-noise, $\boldsymbol{\xi}_{\mathrm{a}}$, while the outer expectation and outer variance samples initial states, $\mathbf{s}$, and action-means, $\boldsymbol{\mu}_{\mathrm{a}}$. To relate this expression to Taylor TD, recall that Taylor TD updates are motivated as performing analytic integration over action-noise, $\boldsymbol{\xi}_{\mathrm{a}}$ - i.e. $\Delta_{\text{Exp}}\theta(\mathbf{s},\boldsymbol{\mu}_{\mathrm{a}}) = \text{E}_{\boldsymbol{\xi}_{\mathrm{a}}}[\Delta\theta|\mathbf{s},\boldsymbol{\mu}_{\mathrm{a}}]$. Thus, the variance of standard (sample-based) TD-updates (Eq. 20) is exactly the variance of $\Delta_{\text{Exp}}\theta$, plus an additional term to account for variance induced by sampling action-noise,

$$\text{Var}_{\mathbf{s},\mathbf{a}}[\Delta\theta(\mathbf{s},\mathbf{a})] = \text{E}_{\mathbf{s},\boldsymbol{\mu}_{\mathrm{a}}}\left[\text{Var}_{\boldsymbol{\xi}_{\mathrm{a}}}[\Delta\theta(\mathbf{s},\boldsymbol{\mu}_{\mathrm{a}}+\boldsymbol{\xi}_{\mathbf{a}})|\mathbf{s},\boldsymbol{\mu}_{\mathrm{a}}]\right] + \text{Var}_{\mathbf{s},\boldsymbol{\mu}_{\mathrm{a}}}[\Delta_{\text{Exp}}\theta(\mathbf{s},\boldsymbol{\mu}_{\mathrm{a}})] \tag{21}$$

The fact that $\text{E}_{\mathbf{s},\boldsymbol{\mu}_{\mathrm{a}}}\left[\text{Var}_{\boldsymbol{\xi}_{\mathrm{a}}}[\Delta\theta|\mathbf{s},\boldsymbol{\mu}_{\mathrm{a}}]\right]$ is non-negative directly gives a theorem.

**Theorem 3.1.** *The variance for standard (sample-based) TD-updates, $\text{Var}_{\mathbf{s},\boldsymbol{\mu}_a,\boldsymbol{\xi}_a}[\Delta\theta]$, is larger than (or equal to) the variance if we exactly integrate action-noise, $\text{Var}_{\mathbf{s},\boldsymbol{\mu}_a}[\Delta_{Exp}\theta]$,*

$$\text{Var}_{\mathbf{s},\boldsymbol{\mu}_a,\boldsymbol{\xi}_a}[\Delta\theta(\mathbf{s},\mathbf{a})] \geq \text{Var}_{\mathbf{s},\boldsymbol{\mu}_a}[\Delta_{Exp}\theta(\mathbf{s},\boldsymbol{\mu}_a)] \tag{22}$$

While this theorem only formally applies to the exact expectation, $\Delta_{\text{Exp}}\theta$, we would expect it to become increasingly accurate as $\lambda_{\mathrm{a}}$ and $\lambda_{\mathrm{s}}$ become smaller, and hence the Taylor series approximations become more accurate. Furthermore, in Appendix B we prove that the resulting updates do not affect the stability of (standard) TD-learning even when the Taylor series approximation errors do not vanish (under linear function approximation). Additionally, we show empirical reductions in the update variance for Taylor TD in Sec. 4.1.

## 4 Experiments

### 4.1 Variance reduction

In this section we empirically test the claim that Taylor TD updates are lower variance than standard (sample-based) TD-learning updates. To do so, we take the Taylor TD set-up in Algorithm 1 and perform "batch updates" [12], where given an approximate value function $Q_\theta$ and a policy $\pi$, several $Q_\theta$ updates are computed across several sampled states and actions, updating $Q_\theta$ only once, based on the sum of all updates. Batch updates ensure the variance of the updates is estimated based on the same underlying value function. We compute batch updates for both Taylor TD and standard (sample-based) TD updates, comparing the variance of the updates between the two approaches (see Appendix F for more details). Formally, we aim to (empirically) show that the inequality in Theorem 3.1 (for both states and actions) holds even with the Taylor approximation across a range of standard control tasks,

$$\text{Var}_{\boldsymbol{\mu}_{\mathrm{s}},\boldsymbol{\xi}_{\mathrm{s}},\boldsymbol{\mu}_{\mathrm{a}},\boldsymbol{\xi}_{\mathrm{a}}}[\Delta\theta(\mathbf{s},\mathbf{a})] \geq \text{Var}_{\boldsymbol{\mu}_{\mathrm{s}},\boldsymbol{\mu}_{\mathrm{a}}}[\Delta_{\text{Exp}}\theta(\boldsymbol{\mu}_{\mathrm{s}},\boldsymbol{\mu}_{\mathrm{a}})] \approx \Delta_{\text{Ta}}\theta(\boldsymbol{\mu}_{\mathrm{s}},\boldsymbol{\mu}_{\mathrm{a}}) \tag{23}$$

Fig. (1) shows Taylor TD provides reliably lower variance updates compared to standard TD-learning across all tested tasks, but the Pendulum environment. This finding is consistent with the idea that Taylor TD updates may be most beneficial in higher dimensional state-action spaces, while the Pendulum environment represents the lowest dimensional environment with only four dimensions. This is because sampling estimates (i.e., standard TD-learning) may become less efficient in higher dimensional spaces. We further explore this proposal in a toy example comparing sample-based estimates and Taylor expansions of expected updates. We perform this comparison across data points of different dimensions, and find the benefits of the Taylor expansion (over purely sample-based estimate) to increase with the dimension of the data points (see Appendix H). Additionally, in the Appendix G, we investigate how the variance of Taylor TD and standard TD-learning updates changes during training.

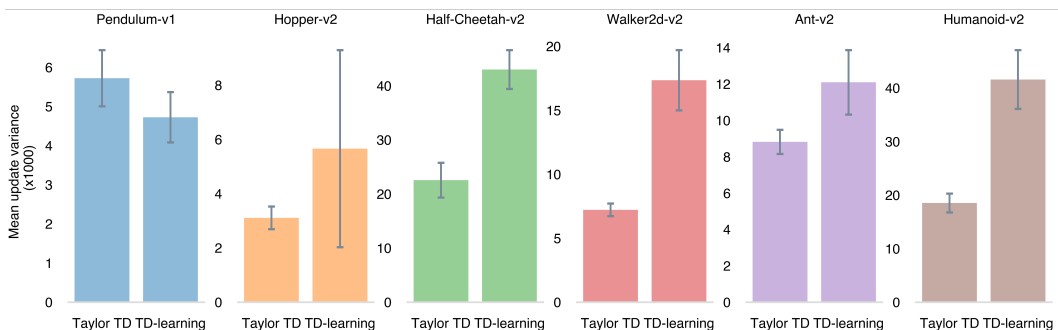

Figure 1: Mean update variance (and standard error) between Taylor TD and standard (sample-based) TD-learning (batch) updates, based on several sampled states and the distribution of actions for those states (i.e. the policy). All results are based on 10 runs with corresponding error bars.

## 4.2 Benchmark performance

### 4.2.1 Algorithm

We combine Taylor TD (i.e. Algorithm 1) with the TD3 algorithm [1] in a model-based off-policy algorithm we call Taylor TD3 (TaTD3) (code available at Appendix I). TaTD3 aims to provide a state-of-the-art implementation of Taylor TD for comparison with baseline algorithms. At each iteration, TaTD3 uses a learned model of the transitions and learned reward function to generate several differentiable (imaginary) 1-step transitions (i.e., Dyna style), starting from real states sampled from a reply buffer. These differentiable 1-step transitions are then used to train two critics (i.e. TD3) using Taylor TD updates. The model of the transitions consists of an ensemble of 8 models trained by maximum likelihood on the observed environment transitions (see Appendix E). These models return a Gaussian distribution over the next state, with zero correlations, and with mean and variance given by applying a neural network to the previous state-action pair. We also learn a model the rewards using a neural network trained with mean-square error on the observed rewards. Hence, TaTD3 does not require any a priori knowledge of the true environment transitions or true reward function. Finally, the actor is trained with the deterministic policy gradient [7] on real states as in standard TD3 [1].

### 4.2.2 Environments

We employ 6 standard environments for continuous control. The first environment consists of a classic problem in control theory used to evaluate RL algorithms [i.e. Pendulum, 27]. The other 5 environments are stanard MuJoCo continous control tasks [i.e. Hopper, HalfCheetah, Walker2d, Ant and Humanoid, 28]. All results are reported in terms of average performance across 5 runs, each with a different random seed (shade represents 95% CI).

### 4.2.3 Comparison with baselines

Here, we report the comparison of TaTD3 with some state-of-the art model -free and -based baselines on the six benchmark environments. These baselines include 3 model-based algorithms and one 1 model-free algorithm (see Appendix J). The first model-based algorithm is Model-based Policy Optimization (MBPO) [18], which employs the soft actor-critic algorithm (SAC) [2] within a model-based Dyna setting. The second model-based algorithm is Model-based Action-Gradient-Estimator Policy Optimization (MAGE) [19], which uses a differentiable model of the environment transitions to train the critic by minimising the norm of the action-gradient of the TD-error. The third model-based algorithm is TD3 combined with a model-based Dyna approach (i.e. Dyna-TD3). This algorithm was proposed by [19] and was shown to outperform its model-free counterpart, TD3 [1] on most benchmark tasks. Dyna-TD3 is conceptually similar to MBPO, with the main difference of MBPO relying on SAC instead of TD3. Finally, we included SAC [2] as a model-free baseline. Fig. (2) shows TaTD3 performs at least as well, if not better, than the baseline algorithms in all six benchmark tasks: note the much poorer performance of MAGE on Hopper-v2, Walker2d-v2 and Ant-v2, as well as of MBPO on Humanoid-v2 relative to TaTD3.

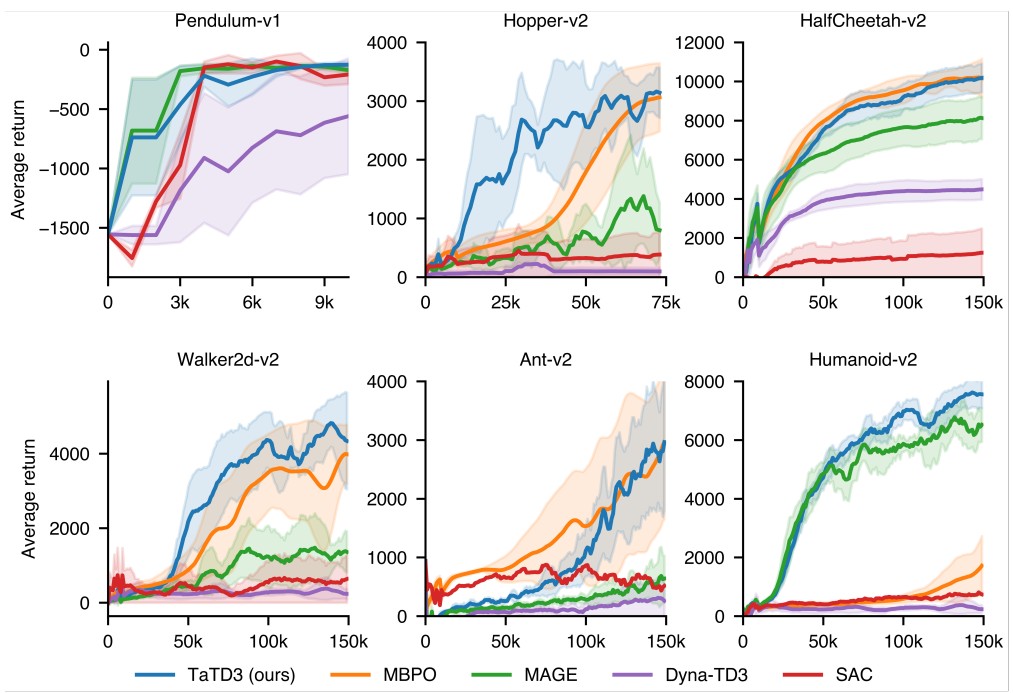

Figure 2: Performance in terms of average returns for TaTD3 and four state-of-the-art baseline algorithms on six benchmark continuous control tasks. TaTD3 performs as well, if not better, than the four baseline algorithms on all four tasks. All performance are based on 5 runs, with shade representing 95% c.i.

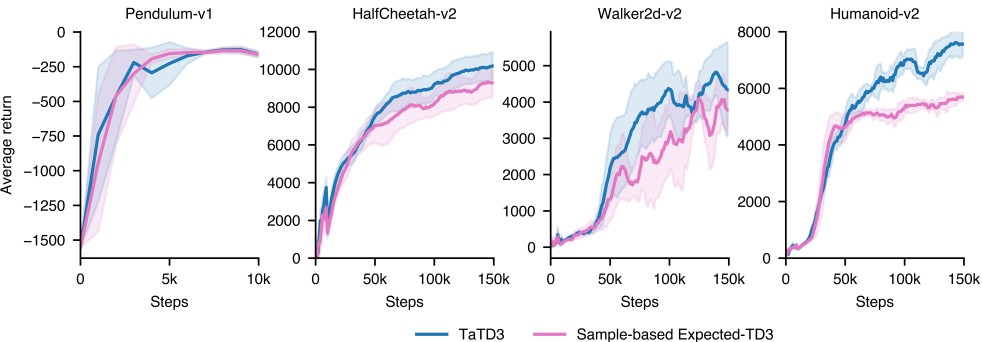

Figure 3: Performance comparison of TaTD3 with its sample-based equivalent, Sample-based Expected-TD3. All performance are based on 5 runs, with shade representing 95% c.i.

#### 4.2.4 Taylor vs sample-based (standard) TD-learning

Next, we ask whether Taylor TD provides any performance benefit in computing the expected TD updates in Eq. (7) and (16) over standard sample-based estimates. To do so, we implement a model-based TD3 algorithm analogous to TaTD3, but where the expected TD updates, Eq. (7) and (16), are estimated using sample-based estimates by drawing multiple samples for the state ($\xi_s$) and action ($\xi_a$) noise at each step, instead of using an analytic Taylor series approximation. We call this algorithm Sample-based Expected-TD3 (code available at Appendix I). In practice, at each time step, Sample-based Expected-TD3 uses a (learned) model of the transitions to compute multiple TD-updates by sampling ten state perturbations ($\xi_s$) of visited states ($\mu_s$) and ten action perturbations ($\xi_a$) of the deterministic target policy ($\mu_a$)(i.e. estimating Eq. (7) and (16) through a sample-based estimate). Crucially, we ensure the variance of the state and action perturbations

(i.e. $\lambda_a$ and $\lambda_s$) is matched between TaTD3 and Sample-based Expected-TD3. We perform this comparison across 4 standard continuous control tasks, spanning a low dimensional (Pendulum), two moderate dimensional (Half-Cheetah and Walker2d) and one high dimensional environment (Humanoid).

In Fig. (3), we can see TaTD3 provides performance benefits over Sample-based Expected-TD3 across the three most high dimensional environments. This finding is in line with the claim we put forth in the variance reduction section (i.e., 4.1) that the benefits of Taylor TD (i.e TaTD3) over sampling may be most marked in high dimensional state-action spaces. Indeed, the largest performance advantage of TaTD3 is seen in Humanoid-v2, which has the highest dimensional state-action space by a large margin. Conversely, the least difference between TaTD3 and Sample-based Expected-TD3 is seen in Pendulum-v1, which is the task with smallest dimensional state-action space. The relation between the variance of TD-updates and the dimension of the state-action space should be better explored in the future (see also Appendix H). Finally, we should stress Sample-based Expected-TD3 is different from Dyna-TD3, as the latter does not implement any action or state perturbation in the TD-updates. Hence, unlike Sample-based Expected-TD3, Dyna-TD3 does not directly estimate the expected updates in Eq. (7) and (16), but relies on (standard) TD-learning updates (this is also evident in the massive performance difference between Dyna-TD3 and Sample-based Expected-TD3 - by comparing corresponding performance between Fig. 2 and 3).

## 5    Related work

Model-based strategies provide a promising solution to improving the sample complexity of RL algorithms [14]. In Dyna methods, a model of the environment transitions is learned through interactions with the environment and then employed to generate additional imaginary transitions [e.g. in the form of model roll-outs, 15]. These imaginary transitions, can be used to enhance existing model-free algorithms, leading to improved sample complexity. For example, within TD-learning, imaginary transitions can be used to train a Q-function by providing additional training examples [e.g. 15, 16, 19]. Alternatively, imaginary transitions can be used to provide better TD targets for existing data points [e.g. 17] or to train the actor and/or critic by generating short-horizon trajectories starting at existing state-action pairs [e.g. 18, 29, 20]. These (Dyna) approaches have a clear relation to our approach (Taylor TD), as they attempt to estimate a similar expected TD-update as in Eq. (7). However, Dyna approaches only use potentially high-variance sample-based estimates, while Taylor TD exploits analytic results to reduce that variance.

Conceptually, our approach may resemble previous methods that also rely on analytical computations of expected updates to achieve lower-variance critic or policy updates [e.g. 24, 30, 31, 12, 32, 33, 34] [see also 35, 36, 37, 38, for a different set of approaches relating to update variance in RL]. The most well-known example of this is Expected-SARSA. Expected-SARSA achieves a lower variance TD-update (relative to SARSA), by analytically computing the expectation over the distribution of target actions in the TD-update (i.e. assuming a stochastic target policy) [31, 12];

$$\delta_\theta(\mathbf{s}, \mathbf{a}) = r(\mathbf{s}, \mathbf{a}) + \gamma \, \mathrm{E}_{a' \sim \pi} \left[ Q_\theta(\mathbf{s}', \mathbf{a}') \right] - Q_\theta(\mathbf{s}, \mathbf{a}) \tag{24}$$

This approach can only reduce variance of TD-updates at the level of the target actions, $\mathbf{a}'$, induced by a stochastic target policy. In the case of a deterministic target policy, Expected-SARSA does not provide any benefit. Conversely, our approach attempts to reduce the variance at the level of the initial state-action pairs, $(\mathbf{s}, \mathbf{a})$ at which TD-updates are performed. That is; we take the expectation over $(\mathbf{s}, \mathbf{a})$ instead of $\mathbf{a}'$ (see Eq. 7 and 16) which should yield benefits with both stochastic and deterministic target policies. Other well-known RL approaches exploiting analytical computations of expected updates are Expected Policy Gradients [24, 30, 33], Mean Actor Critic [32] and "all-action" policy gradient [34]. These methods attempt to reduce the variance of the stochastic policy gradient update by integrating over the action distribution. Although similar in principle to our approach, these methods focus on the policy update instead of the critic update and, similarly to Expected-SARSA only apply to stochastic target policies.

In practice, our approach may relate to value gradient methods, as it explicitly incorporates the gradient of the value function into the update [e.g. 39, 29, 40, 19, 41, 42]. To our knowledge, the value gradient approach that most relates to our work is MAGE [19], which, nonetheless, has a radically different motivation from Taylor TD. MAGE is motivated by noting that the action-gradients of Q drive deterministic policy updates [7], so getting the action-gradients right is critical for policy

learning. In order to encourage the action-gradients of Q to be correct, MAGE explicitly adds a term to the objective that consists of the norm of the action-gradient of the TD-error, which takes it outside of the standard TD-framework. In contrast, our motivation is to reduce the gradient variance across standard TD updates by performing some analytic integration. We do this through a first-order Taylor expansion of the TD update. This difference in motivation leads to numerous differences in the method and analysis, the least of which is that MAGE uses only the action-gradients, while Taylor TD additionally suggests using the state-gradients, as both the state and action gradients can be used to reduce the variance in the (TD) updates.

The idea of applying a Taylor expansion to learning updates is not new to RL. For instance, [43] uses a (model-free) Taylor expansion of the policy update to provide a formalism connecting trust-region policy search with off-policy correction. This is very different from our approach (Taylor TD), not only because Taylor TD is model-based, but also because we apply the Taylor expansion to the critic instead of the actor update for very different reasons (i.e., reduce the update variance across TD updates). Finally, Taylor TD may be applicable to risk-sensitive RL [e.g., 44, 45, 46, 47]. In the presence of a good model of the transitions, Taylor TD may be able to approximate the values of risky actions (or states) around safe (e.g. deterministic) actions (or visited states), without actually needing to take those actions (or visit those states), thanks to the Taylor expansion of the TD objective.

## 6 Conclusion and Limitations

In this article, we introduce a model-based RL framework, Taylor TD, to help reduce the variance of standard TD-learning updates and, speed-up learning of critics. We theoretically and empirically show Taylor TD updates are lower variance than standard (sample-based) TD-learning updates. We show the extra gradient terms used by Taylor TD do not affect the stable learning guarantees of TD-learning with linear function approximation under a reasonable assumption. Next, we combine Taylor-TD with the TD3 algorithm [1] into a model-based off-policy algorithm we denote as TaTD3. We show TaTD3 performs as well, if not better, than several state-of-the art model-free and model-based baseline algorithms on a set of standard benchmark tasks.

Taylor TD has the limitation that it requires continous state-action spaces as well as a differentiable model of transitions to calculate the additional (TD) gradient terms, i.e. it must be in the model-based rather than model-free setting (see Appendix C). Additionally, the gradient terms in Taylor TD imply additional computational cost; in the Appendix L, we show this cost is not that large in terms of training times and we expect it to reduce as faster automatic differentiation tools are developed [48].

## 7 Acknowledgement

We would like to thank the Wellcome Trust for funding this work as well as Dr. Steward for supporting the purchase of GPU nodes. This work made use of the HPC system Blue Pebble at the University of Bristol, UK.

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

# A First-order Taylor expansion

## A.1 Action expansion update proof

Transposing the scalar inner product, $\boldsymbol{\xi}_a^T \nabla_\mathbf{a} Q_\theta(\mathbf{s}, \boldsymbol{\mu}_a)$, in the first-order Taylor expansion in the main text (Eq. 10) gives,

$$\Delta_{\mathrm{Ta}}\theta(\mathbf{s}, \boldsymbol{\mu}_a) = \mathrm{E}_{\boldsymbol{\xi}_a}\left[\left(\delta_\theta(\mathbf{s}, \boldsymbol{\mu}_a) + \boldsymbol{\xi}_a^T \nabla_\mathbf{a}\delta_\theta(\mathbf{s}, \boldsymbol{\mu}_a)\right) \nabla_\theta\left(Q_\theta(\mathbf{s}, \boldsymbol{\mu}_a) + (\nabla_\mathbf{a} Q_\theta(\mathbf{s}, \boldsymbol{\mu}_a))^T \boldsymbol{\xi}_a\right) \Big| \mathbf{s}, \boldsymbol{\mu}_a\right] \tag{25}$$

As $\nabla_\theta$ is a linear operator, and the action noise, $\boldsymbol{\xi}_a$, does not depend on the policy parameters, $\theta$,

$$\Delta_{\mathrm{Ta}}\theta(\mathbf{s}, \boldsymbol{\mu}_a) = \mathrm{E}\left[\left(\delta_\theta(\mathbf{s}, \boldsymbol{\mu}_a) + \boldsymbol{\xi}_a^T \nabla_\mathbf{a}\delta_\theta(\mathbf{s}, \boldsymbol{\mu}_a)\right)\left(\nabla_\theta Q_\theta(\mathbf{s}, \boldsymbol{\mu}_a) + \nabla_{\theta,\mathbf{a}}^2 Q_\theta(\mathbf{s}, \boldsymbol{\mu}_a)\boldsymbol{\xi}_a\right) \Big| \mathbf{s}, \boldsymbol{\mu}_a\right] \tag{26}$$

where $\nabla_{\mathbf{a},\theta}^2 Q_\theta(\mathbf{s}, \boldsymbol{\mu}_a)$ is a matrix of second derivatives. The expectation of $\boldsymbol{\xi}_a$ is zero, so the terms linear in $\boldsymbol{\xi}_a$ are zero, leading to,

$$\Delta_{\mathrm{Ta}}\theta(\mathbf{s}, \boldsymbol{\mu}_a) = \delta_\theta(\mathbf{s}, \boldsymbol{\mu}_a)\nabla_\theta Q_\theta(\mathbf{s}, \boldsymbol{\mu}_a) + \mathrm{E}_{\boldsymbol{\xi}_a}\left[\left(\boldsymbol{\xi}_a^T \nabla_\mathbf{a}\delta_\theta(\mathbf{s}, \boldsymbol{\mu}_a)\right)\left(\nabla_{\theta,\mathbf{a}}^2 Q_\theta(\mathbf{s}, \boldsymbol{\mu}_a)\boldsymbol{\xi}_a^T\right) | \mathbf{s}, \boldsymbol{\mu}_a\right] \tag{27}$$

Swapping the order of the terms in the expectation (which is valid, as $\left(\boldsymbol{\xi}_a^T \nabla_\mathbf{a}\delta_\theta(\mathbf{s}, \boldsymbol{\mu}_a)\right)$ is a scalar),

$$\Delta_{\mathrm{Ta}}\theta(\mathbf{s}, \boldsymbol{\mu}_a) = \delta_\theta(\mathbf{s}, \boldsymbol{\mu}_a)\nabla_\theta Q_\theta(\mathbf{s}, \boldsymbol{\mu}_a) + \mathrm{E}_{\boldsymbol{\xi}_a}\left[(\nabla_{\theta,\mathbf{a}}^2 Q_\theta(\mathbf{s}, \boldsymbol{\mu}_a)\boldsymbol{\xi}_a)\left(\boldsymbol{\xi}_a^T \nabla_\mathbf{a}\delta_\theta(\mathbf{s}, \boldsymbol{\mu}_a)\right) | \mathbf{s}, \boldsymbol{\mu}_a\right] \tag{28}$$

We can then move the terms independent of $\boldsymbol{\xi}_a$ out of the expectation

$$\Delta_{\mathrm{Ta}}\theta(\mathbf{s}, \boldsymbol{\mu}_a) = \delta_\theta(\mathbf{s}, \boldsymbol{\mu}_a)\nabla_\theta Q_\theta(\mathbf{s}, \boldsymbol{\mu}_a) + \nabla_{\theta,\mathbf{a}}^2 Q_\theta(\mathbf{s}, \boldsymbol{\mu}_a)\, \mathrm{E}_{\boldsymbol{\xi}_a}\left[\boldsymbol{\xi}_a\boldsymbol{\xi}_a^T\right] \nabla_\mathbf{a}\delta_\theta(\mathbf{s}, \boldsymbol{\mu}_a) \tag{29}$$

Finally, we know $\mathrm{E}\left[\boldsymbol{\xi}_a\boldsymbol{\xi}_a^T\right] = \boldsymbol{\Sigma}_a$,

$$\Delta_{\mathrm{Ta}}\theta(\mathbf{s}, \boldsymbol{\mu}_a) = \boldsymbol{\mu}_a)\nabla_\theta Q_\theta(\mathbf{s}, \boldsymbol{\mu}_a) + \nabla_{\theta,\mathbf{a}}^2 Q_\theta(\mathbf{s}, \boldsymbol{\mu}_a)\boldsymbol{\Sigma}_a\nabla_\mathbf{a}\delta_\theta(\mathbf{s}, \boldsymbol{\mu}_a) \tag{30}$$

If we assume the action covariance is isotropic, $\boldsymbol{\Sigma}_a = \lambda_a\mathbf{I}$, we get the (1st-order) Taylor TD-update for the action expansion,

$$\Delta_{\mathrm{Ta}}\theta(\mathbf{s}, \boldsymbol{\mu}_a) = \delta_\theta(\mathbf{s}, \boldsymbol{\mu}_a)\nabla_\theta Q_\theta(\mathbf{s}, \boldsymbol{\mu}_a) + \lambda_a\nabla_{\theta,\mathbf{a}}^2 Q_\theta(\mathbf{s}, \boldsymbol{\mu}_a)\nabla_\mathbf{a}\delta_\theta(\mathbf{s}, \boldsymbol{\mu}_a) \tag{31}$$

## A.2 State expansion update proof

Applying the first-order Taylor expansion to Eq. (16), then following the approach from the previous section,

$$\Delta_{\mathrm{Ta}}\theta(\boldsymbol{\mu}_s, \mathbf{a}) = \mathrm{E}_{\boldsymbol{\xi}_s}\left[\left(\delta_\theta(\boldsymbol{\mu}_s, \mathbf{a}) + \boldsymbol{\xi}_s^T \nabla_\mathbf{s}\delta_\theta(\boldsymbol{\mu}_s, \mathbf{a})\right) \nabla_\theta\left(Q(\boldsymbol{\mu}_s, \mathbf{a}) + (\nabla_\mathbf{s} Q(\boldsymbol{\mu}_s, \mathbf{a}))^T \boldsymbol{\xi}_s\right) | \boldsymbol{\mu}_s, \mathbf{a}\right] \tag{32}$$

$$= \mathrm{E}_{\boldsymbol{\xi}_s}\left[\left(\delta_\theta(\boldsymbol{\mu}_s, \mathbf{a}) + \boldsymbol{\xi}_s^T \nabla_\mathbf{s}\delta_\theta(\boldsymbol{\mu}_s, \mathbf{a})\right)\left(\nabla_\theta Q_\theta(\boldsymbol{\mu}_s, \mathbf{a}) + \nabla_{\theta,\mathbf{s}}^2 Q_\theta(\boldsymbol{\mu}_s, \mathbf{a})\boldsymbol{\xi}_s\right) | \boldsymbol{\mu}_s, \mathbf{a}\right] \tag{33}$$

$$= \delta_\theta(\boldsymbol{\mu}_s, \mathbf{a})\nabla_\theta Q_\theta(\boldsymbol{\mu}_s, \mathbf{a}) + \mathrm{E}_{\boldsymbol{\xi}_s}\left[(\nabla_{\theta,\mathbf{s}}^2 Q_\theta(\boldsymbol{\mu}_s, \mathbf{a})\boldsymbol{\xi}_s)\left(\boldsymbol{\xi}_s^T \nabla_\mathbf{s}\delta_\theta(\boldsymbol{\mu}_s, \mathbf{a})\right) | \boldsymbol{\mu}_s, \mathbf{a}\right] \tag{34}$$

$$= \delta_\theta(\boldsymbol{\mu}_s, \mathbf{a})\nabla_\theta Q_\theta(\boldsymbol{\mu}_s, \mathbf{a}) + \nabla_{\theta,\mathbf{s}}^2 Q_\theta(\boldsymbol{\mu}_s, \mathbf{a})\, \mathrm{E}_{\boldsymbol{\xi}_s}\left[\boldsymbol{\xi}_s\boldsymbol{\xi}_s^T\right] \nabla_\mathbf{s}\delta_\theta(\boldsymbol{\mu}_s, \mathbf{a}) \tag{35}$$

Finally, we know $\mathrm{E}\left[\boldsymbol{\xi}_s\boldsymbol{\xi}_s^T\right] = \boldsymbol{\Sigma}_s$,

$$\Delta_{\mathrm{Ta}}\theta(\boldsymbol{\mu}_s, \mathbf{a}) = \delta_\theta(\boldsymbol{\mu}_s, \mathbf{a})\nabla_\theta Q_\theta(\boldsymbol{\mu}_s, \mathbf{a}) + \nabla_{\theta,\mathbf{s}}^2 Q_\theta(\boldsymbol{\mu}_s, \mathbf{a})\boldsymbol{\Sigma}_s\nabla_\mathbf{s}\delta_\theta(\boldsymbol{\mu}_s, \mathbf{a}) \tag{36}$$

If we assume the state covariance is isotropic, $\boldsymbol{\Sigma}_s = \lambda_s\mathbf{I}$, we get the (1st-order) Taylor TD-update for the state expansion,

$$\Delta_{\mathrm{Ta}}\theta(\boldsymbol{\mu}_s, \mathbf{a}) = \delta_\theta(\boldsymbol{\mu}_s, \mathbf{a})\nabla_\theta Q_\theta(\boldsymbol{\mu}_s, \mathbf{a}) + \lambda_s\nabla_{\theta,\mathbf{s}}^2 Q_\theta(\boldsymbol{\mu}_s, \mathbf{a})\nabla_\mathbf{s}\delta_\theta(\boldsymbol{\mu}_s, \mathbf{a}) \tag{37}$$

# B Proof of stable learning for Taylor TD with linear function approximation (with a fixed policy)

The Taylor TD update for the action expansion can be written as (an equivalent proof can be derived for the state expansion):

$$\Delta_{\text{Ta}}\theta(\mathbf{s}, \boldsymbol{\mu}_{\text{a}}) = \delta_\theta(\mathbf{s}, \boldsymbol{\mu}_{\text{a}})\nabla_\theta Q_\theta(\mathbf{s}, \boldsymbol{\mu}_{\text{a}}) + \lambda_{\text{a}}\nabla^2_{\theta, \boldsymbol{\mu}_{\text{a}}} Q_\theta(\mathbf{s}, \boldsymbol{\mu}_{\text{a}})\nabla_{\mathbf{a}}\delta_\theta(\mathbf{s}, \boldsymbol{\mu}_{\text{a}}) \tag{38}$$

This update is composed of two quantities, the standard TD update (i.e., first term in the sum) plus the extra term induced by the Taylor expansion (i.e., second term in the sum). For the purposes of this proof, we consider linear function approximation,

$$Q_\theta(\mathbf{s}, \boldsymbol{\mu}_{\text{a}}) = \theta^T\mathbf{x} \qquad\qquad Q_\theta(\mathbf{s}', \mathbf{a}') = \theta^T\mathbf{x}' \tag{39}$$

where,

$$\mathbf{x} = \phi(\mathbf{s}, \boldsymbol{\mu}_{\text{a}}) \in \mathbb{R}^N \qquad\qquad \mathbf{x}' = \phi(\mathbf{s}', \mathbf{a}') \in \mathbb{R}^N \tag{40}$$

and where $\mathbf{x}$ and $\mathbf{x}'$ are feature-vectors of length $N$. We can re-write each of the two terms in the Taylor TD update in terms of this linear function approximation. The first term, corresponding to a standard TD-update can be written,

$$\delta_\theta(\mathbf{s}, \mathbf{a})\nabla_\theta Q_\theta(\mathbf{s}, \mathbf{a}) = (r + \gamma\theta^T\mathbf{x}' - \theta^T\mathbf{x})\mathbf{x}$$
$$= r\mathbf{x} - \mathbf{x}(\mathbf{x} - \gamma\mathbf{x}')^T\theta. \tag{41}$$

The second term, which is the new term introduced by Taylor-TD methods, can be written,

$$\nabla^2_{\theta, \mathbf{a}} Q_\theta(\mathbf{s}, \mathbf{a})\nabla_{\mathbf{a}}\delta_\theta(\mathbf{s}, \mathbf{a}) = \nabla_{\theta, \mathbf{a}}\left(\mathbf{x}^T\theta\right)\left(\nabla_{\mathbf{a}}r + \gamma\nabla_{\mathbf{a}}(\mathbf{x}'^T\theta) - \nabla_{\mathbf{a}}(\mathbf{x}^T\theta)\right)$$
$$= (\nabla_{\mathbf{a}}\mathbf{x})^T\left(\nabla_{\mathbf{a}}r + \gamma(\nabla_{\mathbf{a}}\mathbf{x}')\theta - (\nabla_{\mathbf{a}}\mathbf{x})\theta\right)$$
$$= (\nabla_{\mathbf{a}}\mathbf{x})^T\nabla_{\mathbf{a}}r + \gamma(\nabla_{\mathbf{a}}\mathbf{x})^T\nabla_{\mathbf{a}}\mathbf{x}'\theta - (\nabla_{\mathbf{a}}\mathbf{x})^T\nabla_{\mathbf{a}}\mathbf{x}\theta$$
$$= (\nabla_{\mathbf{a}}\mathbf{x})^T\nabla_{\mathbf{a}}r - (\nabla_{\mathbf{a}}\mathbf{x})^T\left(\nabla_{\mathbf{a}}\mathbf{x} - \gamma\nabla_{\mathbf{a}}\mathbf{x}'\right)\theta \tag{42}$$

Here, $\nabla_{\mathbf{a}}\mathbf{x} \in \mathbb{R}^{A \times N}$, is a matrix, while $\nabla_{\mathbf{a}}r \in \mathbb{R}^A$ is a vector.

Putting the two terms together Eq. 41 & 42 and factorising the terms multiplying $\theta_t$, we can write the expected next weight vector as:

$$\mathrm{E}\left[\theta_{t+1} \mid \theta_t\right] = \theta_t + \eta\Delta\theta = (\mathbf{I} - \eta(\mathbf{A} + \lambda_{\text{a}}\tilde{\mathbf{A}}))\theta_t + \eta\mathbf{u} \tag{43}$$

where:

$$\mathbf{u} = \mathrm{E}\left[r\mathbf{x} + (\nabla_{\mathbf{a}}\mathbf{x})^T\nabla_a r\right] \tag{44}$$

$$\mathbf{A} = \mathrm{E}\left[\mathbf{x}(\mathbf{x} - \gamma\mathbf{x}')^T\right] \in \mathbb{R}^{N \times N} \tag{45}$$

$$\tilde{\mathbf{A}} = \mathrm{E}\left[(\nabla_{\mathbf{a}}\mathbf{x})^T\left(\nabla_{\mathbf{a}}\mathbf{x} - \gamma\nabla_{\mathbf{a}}\mathbf{x}'\right)\right] \in \mathbb{R}^{N \times N} \tag{46}$$

Since only $\mathbf{A}$ and $\tilde{\mathbf{A}}$ multiplies $\theta$, these two quantities exclusively are important for guaranteeing stable learning. If $\mathbf{A}$ is positive definite and $\tilde{\mathbf{A}}$ is positive-semi-definite, then $(\mathbf{A} + \lambda_{\text{a}}\tilde{\mathbf{A}})$ is positive-definite, and for sufficiently small $\eta$, the magnitude of the eigenvalues of $\mathbf{I} - \eta(\mathbf{A} + \lambda_{\text{a}}\tilde{\mathbf{A}})$ are all smaller than 1, in which case the system is stable. Crucially, the term $\mathbf{A}$ is the same as traditional TD-learning so [11] provides a proof that $\mathbf{A}$ is always positive definite [see also 12]. Thus, all we have to prove is that $\tilde{\mathbf{A}}$ is positive semi-definite. In order to prove that $\tilde{\mathbf{A}}$ is positive semi-definite, we require an (very reasonable) assumption, that the timestep $\Delta t$ is small. Specifically, if we take,

$$\mathbf{s}' = \Delta t\, f(\mathbf{s}, \mathbf{a}) + \mathbf{s} \tag{47}$$

Then the $\nabla_{\mathbf{a}}\mathbf{x}'$ terms must be small. Specifically, applying the chain rule, the full derivative is (expressed here for 1d case, $\nabla_{\mathbf{a}}\mathbf{x}' = \frac{\partial x'}{\partial a}$, for simplicity),

$$\frac{\partial x'}{\partial a} = \frac{\partial x'}{\partial a'}\frac{\partial a'}{\partial s'}\frac{\partial s'}{\partial a} + \frac{\partial x'}{\partial s'}\frac{\partial s'}{\partial a}\frac{\partial x'}{\partial a} = \left(\frac{\partial x'}{\partial a'}\frac{\partial a'}{\partial s'} + \frac{\partial x'}{\partial s'}\right)\frac{\partial s'}{\partial a} \tag{48}$$

This is proportional to $\Delta t$, as

$$\frac{\partial s'}{\partial a} = \Delta t \frac{\partial f(s, a)}{\partial a} \tag{49}$$

The key test for positive semi-definiteness is that for all vectors $\mathbf{b}$,

$$0 \le \mathbf{b}^T \tilde{\mathbf{A}} \mathbf{b}. \tag{50}$$

Substituting the definition of $\tilde{\mathbf{A}}$ from Eq. (46),

$$0 \le \mathbf{b}^T E[(\nabla_{\mathbf{a}} \mathbf{x})^T \nabla_{\mathbf{a}} \mathbf{x}] \mathbf{b} - \gamma \mathbf{b}^T E[(\nabla_{\mathbf{a}} \mathbf{x})^T \nabla_{\mathbf{a}} \mathbf{x}'] \mathbf{b}. \tag{51}$$

Putting the $\mathbf{b}$'s inside the expectation,

$$0 \le E[(\nabla_{\mathbf{a}} \mathbf{x} \mathbf{b})^T (\nabla_{\mathbf{a}} \mathbf{x} \mathbf{b})] - \gamma E[(\nabla_{\mathbf{a}} \mathbf{x} \mathbf{b})^T (\nabla_{\mathbf{a}} \mathbf{x}' \mathbf{b})] \tag{52}$$

Now, $(\nabla_{\mathbf{a}} \mathbf{x} \mathbf{b}) \in \mathbb{R}^{1 \times A}$ is a length-$A$ row-vector, and the terms inside the expectations are inner products of two length $A$ vectors. We can write these vector inner products as sums,

$$0 \le \sum_{i=1}^{A} E[(\nabla_{a_i} \mathbf{x} \mathbf{b})^T (\nabla_{a_i} \mathbf{x} \mathbf{b})] - \gamma E[(\nabla_{a_i} \mathbf{x} \mathbf{b})^T (\nabla_{a_i} \mathbf{x}' \mathbf{b})] \tag{53}$$

where $a_i$ is a particular element of the action-vector, $\mathbf{a}$, so $\nabla_{a_i} \mathbf{x} \in \mathbb{R}^N$ is a length-N vector, and

$$\nabla_{\mathbf{a}} \mathbf{x} = \begin{pmatrix} \nabla_{a_1} \mathbf{x} \\ \nabla_{a_2} \mathbf{x} \\ \vdots \\ \nabla_{a_A} \mathbf{x} \end{pmatrix} \tag{54}$$

Critically, the overall inequality in Eq. (52) holds if a similar inequality holds for every term in the sum in Eq. (53),

$$0 \le E[(\nabla_{a_i} \mathbf{x} \mathbf{b})^T (\nabla_{a_i} \mathbf{x} \mathbf{b})] - \gamma E[(\nabla_{a_i} \mathbf{x} \mathbf{b})^T (\nabla_{a_i} \mathbf{x}' \mathbf{b})] \tag{55}$$

As $\nabla_{a_i} \mathbf{x} \mathbf{b}$ and $\nabla_{a_i} \mathbf{x}' \mathbf{b}$ are scalars, we can write,

$$0 \le E[(\nabla_{a_i} \mathbf{x} \mathbf{b})^2] - \gamma E[(\nabla_{a_i} \mathbf{x} \mathbf{b})(\nabla_{a_i} \mathbf{x}' \mathbf{b})] \tag{56}$$

There are now two cases. If $0 = E[(\nabla_{a_i} \mathbf{x} \mathbf{b})^2]$, we must have that $0 = (\nabla_{a_i} \mathbf{x} \mathbf{b})$ always (except at a set of measure zero). Thus, the second term must also be zero (except at a set of measure zero), i.e. $0 = E[(\nabla_{a_i} \mathbf{x} \mathbf{b})(\nabla_{a_i} \mathbf{x}' \mathbf{b})]$ and the inequality holds. Alternatively, if $E[(\nabla_{a_i} \mathbf{x} \mathbf{b})^2]$ is non-zero, it must be positive, in that case, the second term can also be non-zero, but it scales with $\Delta t$, so we can always choose a $\Delta t$, small enough to ensure that the Eq. (56) holds, in which case $\tilde{\mathbf{A}}$ is positive semi-definite.

## C    Using the chain rule to expand the gradient of Taylor TD

A (learned) differentiable model of the transitions and rewards is needed to compute the gradient terms $\nabla_{\mathbf{a}} \delta_\theta(\mathbf{s}, \mathbf{a})$ and $\nabla_{\mathbf{s}} \delta_\theta(\mathbf{s}, \mathbf{a})$ in Taylor TD. This is because the TD target, $\delta_\theta = r(\mathbf{s}, \mathbf{a}) + \gamma Q_\theta(\mathbf{s}', \mathbf{a}')$, comprises the reward and the Q-value at the next time step, both of which depend on $\mathbf{s}$ and $\mathbf{a}$. The full gradients for the action expansion can be written:

$$\nabla_{\mathbf{a}} \delta_\theta(\mathbf{s}, \mathbf{a}) = \frac{\partial \hat{r}(\mathbf{s}, \mathbf{a})}{\partial \mathbf{a}} - \frac{\partial Q_\theta(\mathbf{s}, \mathbf{a}_t)}{\partial \mathbf{a}} + \gamma \frac{\partial \hat{\mathbf{s}}'}{\partial \mathbf{a}} \left( \frac{\partial Q_\theta(\hat{\mathbf{s}}', \mathbf{a}')}{\partial \hat{\mathbf{s}}'} + \frac{\partial \mathbf{a}'}{\partial \hat{\mathbf{s}}'} \frac{\partial Q_\theta(\hat{\mathbf{s}}', \mathbf{a}')}{\partial \mathbf{a}'} \right) \tag{57}$$

where $\hat{r}$ denotes a (differentiable) reward function, $\hat{s}$ denotes the predicted next state, while $\frac{\partial \hat{\mathbf{s}}'}{\partial \mathbf{a}}$ denotes the gradient term computed by differentiating through a differentiable model of the transitions. An analogous gradient can be written for $\nabla_{\mathbf{s}} \delta_\theta(\mathbf{s}, \mathbf{a})$ (i.e., state expansion). However, we do not have to explicitly implement these expressions, as we use autodiff in PyTorch to find gradients of $\delta_\theta(\mathbf{s}, \mathbf{a})$ wrt $\mathbf{a}$ and $\mathbf{s}$ directly, by re-writing the Taylor TD updates in terms of a simple loss (see Appendix D).

## D  Computing the updates as the gradient of a loss

Here we report how to easily implement the Taylor TD-update (i.e. Eq. 18) as a loss to be passed to an optimizer (e.g. PyTorch optimizer).

$$\mathcal{L}_\theta^{\text{critic}} = - \ \text{stopgrad}_\theta\{\delta_\theta(\boldsymbol{\mu}_{\text{s}}, \boldsymbol{\mu}_{\text{a}})\} \ Q_\theta(\boldsymbol{\mu}_{\text{s}}, \boldsymbol{\mu}_{\text{a}})$$
$$- \lambda_{\text{a}} \frac{\text{stopgrad}_\theta\{\nabla_{\mathbf{a}}\delta_\theta(\boldsymbol{\mu}_{\text{s}}, \boldsymbol{\mu}_{\text{a}})\} \cdot \nabla_{\mathbf{a}} Q_\theta(\boldsymbol{\mu}_{\text{s}}, \boldsymbol{\mu}_{\text{a}})}{\text{stopgrad}_\theta\{\| \left( \nabla_{\mathbf{a}}\delta_\theta(\boldsymbol{\mu}_{\text{s}}, \boldsymbol{\mu}_{\text{a}}) \right) \| \| \nabla_{\mathbf{a}} Q_\theta(\boldsymbol{\mu}_{\text{s}}, \boldsymbol{\mu}_{\text{a}}) \| \}}$$
$$- \lambda_{\text{s}} \frac{\text{stopgrad}_\theta\{(\nabla_{\mathbf{s}}\delta_\theta(\boldsymbol{\mu}_{\text{s}}, \boldsymbol{\mu}_{\text{a}}))\} \cdot \nabla_{\mathbf{s}} Q_\theta(\boldsymbol{\mu}_{\text{s}}, \boldsymbol{\mu}_{\text{a}})}{\text{stopgrad}_\theta\{\| \left( \nabla_{\mathbf{s}}\delta_\theta(\boldsymbol{\mu}_{\text{s}}, \boldsymbol{\mu}_{\text{a}}) \right) \| \| \nabla_{\mathbf{s}} Q_\theta(\boldsymbol{\mu}_{\text{s}}, \boldsymbol{\mu}_{\text{a}}) \| \}}$$

"$\text{stopgrad}_\theta\{\}$" denotes the optimizer should not differentiate the quantity inside the curly brackets relative to the parameters $\theta$ (i.e. equivalent to ".detach()" in PyTorch).

## E  Model learning

Taylor TD-learning requires a differentiable model of the transitions. We learned this model according to maximum-likelihood based on observed transitions sampled from a reply buffer,

$$\mathcal{L}_w^{\text{model}} = \hat{p}_w(s' \mid s, a); \qquad\qquad (s, a, s') \sim \mathcal{B} \qquad\qquad (58)$$

where $\mathcal{B}$ denotes a reply buffer from which (observed) transitions can be sampled. In TaTD3, the model of the transitions consists of an ensemble of 8 models. These models return a Gaussian distribution over the next state, with zero correlations, and with mean and variance given by applying a neural network to the previous state-action pair. Conversely, the model of the rewards consists of a neural network trained with mean-square error on the observed rewards sampled from the reply buffer.

## F  Variance reduction

Here we describe in more details how we computed the variance of Taylor TD and standard TD-learning updates on the 6 continuous control benchmark tasks (i.e. Section 4.1). Based on a policy, a value function and a set of randomly sampled states from a memory buffer, we computed the Taylor and the standard (sample-based) TD batch updates across all sampled states (under the policy). Note, we ensured the exact same (random) states were used to compute the Taylor and the standard (sample-based) TD batch updates. The update for each state takes the form of a gradient evaluation of the value function relative to the TD objective. Next, we computed the variance of these gradient terms across all states and summed them up since the variance of each parameter update contributes to the variance of the value function relative to the TD objective. We repeated this process for 10 different seeds and plotted the mean update variance and standard error for both Taylor and the standard TD updates (i.e. Fig.1).

## G  Variance reduction across training

In this section we test how the variance of Taylor TD and standard TD-learning updates changes across training. In particular, we want to assess how the variance advantage of Taylor TD updates (over standard TD updates) changes across 3 different points in training: at the start of training (i.e., for an untrained critic and an untrained model of the transitions), early in training (i.e., after 5000 training steps) and finally late in training (i.e., after 50000 training steps).

In Fig. 4, we can see that across all training points, Taylor TD updates provide lower variance updates compared to standard TD updates.

## H  Toy example

We provide a toy example to investigate the settings in which using a Taylor expansion of an expected update may provide the largest benefits relative to sample-based estimates of the same expected

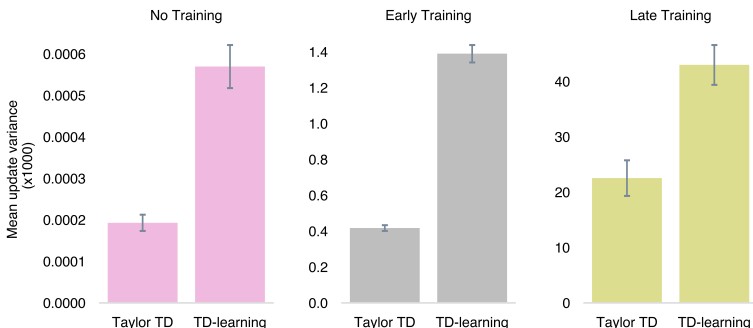

Figure 4: Mean update variance (and standard error) between Taylor TD and standard (sample-based) TD-learning (batch) updates, computed across 3 time points in training: No Training (i.e., for an untrained critic), Early training (i.e., after 5000 training steps) and finally late in training (i.e., after 50000 training steps) . All results are based on 10 runs with corresponding error bars on the Half-Cheetah-v2 environment.

update. To do so, we train a function approximation parameterised by $\theta$ (e.g. a critic) to approximate the expected outputs of an underlying target function (e.g. TD targets). Starting from a single expected target value, we can formulate the following objective (for multiple targets, we can just sum the objectives):

$$J(\theta) = \frac{1}{2} \mathrm{E}_{\boldsymbol{\xi}_\mathrm{x}} \left[ (y(x + \boldsymbol{\xi}_\mathrm{x}) - \hat{y}_\theta(x + \boldsymbol{\xi}_\mathrm{x}))^2 \right] \tag{59}$$

where $\boldsymbol{\xi}_\mathrm{x}$ is sampled form a Gaussian distribution with $\mathrm{E}\left[\boldsymbol{\xi}_\mathrm{x}\right] = \mathbf{0}$, $\mathrm{E}\left[\boldsymbol{\xi}_\mathrm{x}\boldsymbol{\xi}_\mathrm{x}^T\right] = \lambda_\mathrm{x}\mathbf{I}$. Hence, the task requires the function approximation $\hat{y}_\theta()$ to approximate the expected outputs of the target function $y()$, based on a set of randomly perturbed inputs (i.e. $x + \boldsymbol{\xi}_\mathrm{x}$). We do this by comparing two different approaches. The first approach involves sampling different outputs of $y$ (based on different input perturbations $\boldsymbol{\xi}_\mathrm{x}$) and training $\hat{y}_\theta()$ to match those outputs. We denote this approach as "Sample-based targets". This approach aims to mimic standard TD-learning, where TD-targets are sampled at each time step to train the critic (i.e. providing sample-based estimates). The second approach aims to mimic Taylor TD, applying a first-order Taylor expansion to the objective in Eq 59,

$$J_{\mathrm{Ta}_{\boldsymbol{\xi}_\mathrm{a}}}(\theta) = \frac{1}{2} \mathrm{E}_{\boldsymbol{\xi}_\mathrm{x}} \left[ (y + \boldsymbol{\xi}_\mathrm{x}^T \nabla_\mathrm{x} y - \hat{y}_\theta - \boldsymbol{\xi}_\mathrm{x}^T \nabla_\mathrm{x} \hat{y}_\theta)^2 \right] \tag{60}$$

For clarity we used the notation $\hat{y}_\theta = \hat{y}_\theta(x)$ and $y = y(x)$. The terms that are linear in $\boldsymbol{\xi}_\mathrm{x}$ cancels and after summing up equal terms, we get:

$$J_{\mathrm{Ta}_{\boldsymbol{\xi}_\mathrm{a}}}(\theta) = \frac{1}{2}(y - \hat{y}_\theta)^2 + \frac{1}{2}\lambda_\mathrm{x}(\nabla_x y - \nabla_x \hat{y}_\theta)^T(\nabla_x y - \nabla_x \hat{y}_\theta) \tag{61}$$

We can then take the gradient of this approximation relative to the function approximation weights to get the weight update for the Taylor approach:

$$\begin{aligned} \nabla_\theta J(\theta) \sim\ & \hat{y}_\theta \nabla_\theta \hat{y}_\theta \\ & + \lambda_\mathrm{x} \nabla_\theta (\nabla_x \hat{y}_\theta^T \nabla_x \hat{y}_\theta) \\ & - y \nabla_\theta \hat{y}_\theta \\ & - \lambda_\mathrm{x} \nabla_\theta (\nabla_x y^T \nabla_x \hat{y}_\theta^T) \end{aligned} \tag{62}$$

Note, this update is reminiscent of of double backpropagation settings in the supervised learning literature [49, 50]. The toy example allows us to compare the Taylor expansion with the sample-based estimation under different conditions (i.e. dimension size of the data points as well as number of data points). In particular, we compare the two approaches for inputs of different dimensions (from 1 to 100 dimensional inputs) and across two data regimes, a low data regime (i.e. only 15 training samples are used to train the function approximation) and a high data regime (i.e. over 100 samples

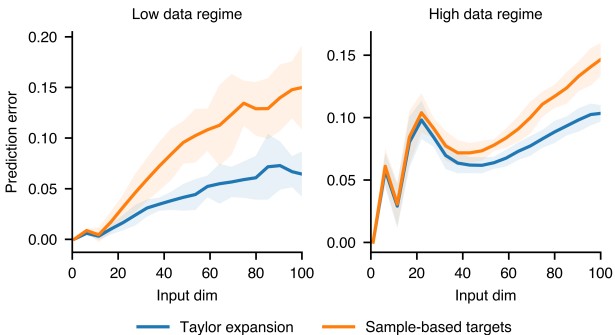

Figure 5: Average performance of the Taylor expansion approach relative to sample-based estimates on unseen input examples across several input dimensions (x-axis) and two data regimes (5 runs, 95% c.i.).

are used to train the function approximation). Performance are then assessed based on a novel set of inputs (i.e. 50), sampled from the same underlying distribution as the training data. Fig. 5 show that the benefits of the Taylor approach over sample-based estimates increase as the dimension of the data points grows in size (e.g. RL tasks involving high dimensional action and state spaces). In particular, these benefits are even larger in the presence of a low data regime, such as RL settings in which for any given state-action pair we can only sample a few transitions from the environment.

## I  Code

All the code is available at `https://github.com/maximerobeyns/taylortd`

## J  Baseline algorithms

Plotted performance of MBPO [18] was directly taken from the official algorithm repository on GitHub at `https://github.com/jannerm/mbpo`. Plotted performance of SAC [2] was also obtained taken from the official algorithm repository on GitHub at `https://github.com/haarnoja/sac`. Plotted performances of both MAGE and Dyna-TD3 [19] were obtained by re-running these algorithms on the benchmark environments, taking the implementations from the official algorithms' repository available at `https://github.com/nnaisense/MAGE`.

## K  Ablations

### K.1  State expansion ablation

Here, we ask whether the Taylor state expansion brings any benefit to performance, on top of the Taylor action expansion. To do so, we compare the TaTD3 algorithms with and without state expansion on two standard benchmark tasks (i.e. analogous to setting $\lambda_s = 0$ in the update Eq. 18). Fig. (6) shows that including the state expansion is beneficial to both environments.

### K.2  Cosine similarity ablation

Here, we ask whether taking the cosine similarity of state and action gradient terms benefit the performance of TaTD3. To do so, we compare the standard TaTD3 algorithms (trained with the loss in Eq. 18) with a version of TaTD3 trained with a loss without cosine similarity,

$$\mathcal{L}_\theta = \delta(\boldsymbol{\mu}_s, \boldsymbol{\mu}_a) Q_\theta(\boldsymbol{\mu}_s, \boldsymbol{\mu}_a) + \lambda_a \, \nabla_{\mathbf{a}} Q_\theta(\boldsymbol{\mu}_s, \boldsymbol{\mu}_a) \cdot \nabla_{\mathbf{a}} \delta(\boldsymbol{\mu}_s, \boldsymbol{\mu}_a) + \lambda_s \, \nabla_{\mathbf{s}} Q_\theta(\boldsymbol{\mu}_s, \boldsymbol{\mu}_a) \cdot \nabla_{\mathbf{s}} \delta(\boldsymbol{\mu}_s, \boldsymbol{\mu}_a)$$
(63)

Namely, this loss optimizes the dot-product for state and action gradient terms instead of the cosine similarity. We assess this comparison on two standard benchmark tasks (i.e. see Fig. 7). In Fig. 7, we can see the cosine similarity does improve performance on both tasks.

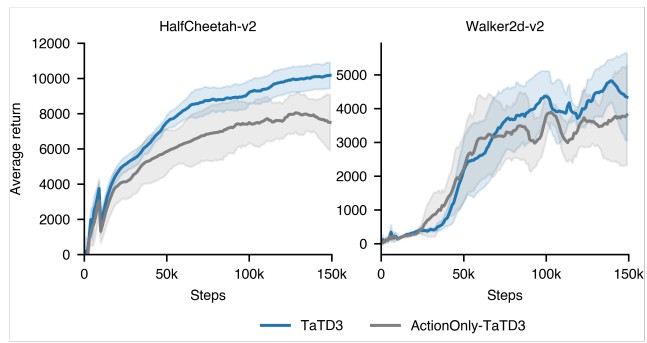

Figure 6: Performance in terms of average returns for TaTD3 (with both state and action expansions) compared to a version of TaTD3 that uses the action expansion only, on two benchmark continuous control tasks. Including the state expansion in TaTD3 seem to improve performance on both tasks (5 runs, 95% c.i.).

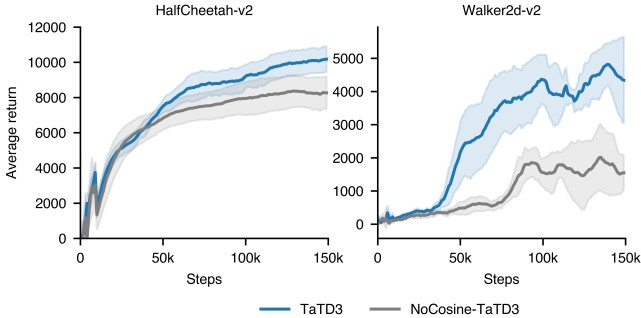

Figure 7: Performance in terms of average returns for TaTD3 compared to a version of TaTD3 that computes the dot product of state and action gradient terms, instead of cosine similarity (5 runs, 95% c.i.).

## L   Computing

All experiments were run on a cluster of GPUs, including NVIDIA GeForce RTX 2080, 3090 and NDVIDIA A100. Here, we report the difference in computing time between standard TD-learning, Taylor TD and the two strongest baselines (i.e., MAGE and MBPO) based on the same GPU for each comparison. We do so for a low dimensional (Pendulum), a 'medium' dimensional (Walker2d) and a high dimensional (Humanoid) environment to span a broad range of settings. For MBPO, we report the average running time across all training epochs using the horizon lengthening schedule outlined in the original paper [18].

| Environment | TD-learning time | Taylor TD time | MAGE time | MBPO time | n. time steps |
|---|---|---|---|---|---|
| Pendulum | 24s | 38s | 36s | 52s | 200 |
| Walker2d | 50s | 68s | 63s | 133s | 1000 |
| Humanoid | 94s | 117s | 127s | 235s | 1000 |

MAGE has about the same runtime as our method (TaTD3), while MBPO is a lot slower than our method.

## M   Hyperparameters settings

Below, we reported the hyperparameter settings for TaTD3 (and sample-based Expected-TD3),

|  | Pendulum-v1 | HalfCheetah-v2 | Walker2d-v2 |
|---|---|---|---|
| Steps | 10000 | 150000 | 150000 |
| Model ensemble size | 8 | 8 | 8 |
| Model architecture (MLP) | 4 h-layers of size 512 | 4 h-layers of size 512 | 4 h-layers of size 512 |
| Reward model architecture (MLP) | 3 h-layers of size 256 | 3 h-layers of size 256 | 3 h-layers of size 256 |
| Actor-critic architecture (MLP) | 2 h-layers of size 400 | 2 h-layers of size 400 | 2 h-layers of size 400 |
| Dyna steps per environment step | 10 | 10 | 10 |
| Model horizon | 1 | 1 | 1 |
| $\lambda_a$ | 0.25 | 0.25 | 0.25 |
| $\lambda_s$ | 1e-5 | 1e-5 | 1e-5 |

|  | Hopper-v2 | Ant-v2 | Humanoid-v2 |
|---|---|---|---|
| Steps | 10000 | 150000 | 150000 |
| Model ensemble size | 8 | 8 | 8 |
| Model architecture (MLP) | 4 h-layers of size 512 | 4 h-layers of size 512 | 4 h-layers of size 512 |
| Reward model architecture (MLP) | 3 h-layers of size 256 | 3 h-layers of size 512 | 3 h-layers of size 512 |
| Actor-critic architecture (MLP) | 2 h-layers of size 400 | 4 h-layers of size 400 | 4 h-layers of size 400 |
| Dyna steps per environment step | 10 | 10 | 10 |
| Model horizon | 1 | 1 | 1 |
| $\lambda_a$ | 0.06 | 0.06 | 0.25 |
| $\lambda_s$ | 1e-5 | 1e-5 | 1e-5 |

Note, "h-layers" stands for hidden layers and the size is in terms of number of units.

We found we could achieve good performance for TaTD3 across all tested environments without needing to fine tune the value of $\lambda_a$ and $\lambda_s$ to each environment (i.e., $\lambda_a = 0.06$ and $\lambda_s = 0.00005$). These parameters were founded by running a grid search over potential values of these parameters based on a single environment (i.e., Pendulum), then using the best values for all other environments. Nevertheless, we reached top performance in HalfCheetah, Walker2d and Humanoid by using a larger $\lambda_a$ (i.e., $\lambda_a = 0.25$). This finding suggests these 3 environments benefit from learning a broader distribution of Q-values over the actions, we believe for better exploration. Additionally, you may notice the much larger values for $\lambda_a$ compared to $\lambda_s$. However, we should stress that any direct comparison between the state covariance $\lambda_s$ and the action covariance $\lambda_a$ may not be very meaningful. This is because the distribution of initial states and the distribution of initial actions may be very different, making any direct comparison between $\lambda_s$ and $\lambda_a$ hard to assess.

