# OpenReview forum: "Taylor TD-learning"
_NeurIPS.cc/2023/Conference — NeurIPS 2023 poster_

### Official Review · Reviewer_9ew5 · 2023-07-06

**Soundness:** 3 good
**Presentation:** 3 good
**Contribution:** 3 good
**Rating:** 7
**Confidence:** 4

**Summary:**

This paper proposes an algorithm for improving model-based actor-critic methods in continuous state and action spaces. When using Dyna-style updates, the algorithm can compute an expected update over a small noise distribution using a linearization to reduce the variance of the performed update. This method is shown to be theoretically-sound in a simplified setting and a variety of experiments demonstrate the effectiveness of the proposed approach on standard continuous-control benchmark tasks.


**Strengths:**

- The main idea is interesting, computing an explicit expected update over a small region by using a linearization.
- The experiments are done well with meaningful baseline algorithms. In particular, the supporting experiments other than standard learning curves are a welcome addition: The variance analysis comparing sampled updates to the proposed one and the ablation study on cosine similarity vs. the inner product (in the appendix).
- The theoretical result in the linear setting is nice even if it is in a simplified setting and the proofs are correct as far as I can tell.



**Weaknesses:**

There are some aspects of the presentation that I think could be improved. These are more minor points overall though.
- For example, I would avoid using the term "Monte-Carlo" estimates when discussing sampling of TD updates since it can lead to some confusion due to "Monte-Carlo" often being used to refer to MC estimates of the return in contrast to the bootstrap estimates that TD uses. I would consider using the phrase "sampled TD update" instead of "MC TD update". E.g. line 31, line 42, ...
- I would also suggest including a line about continuous state and actions in the introduction to clarify the problem setting since the proposed algorithm would mainly be applicable there.
- Section 3.1 describing the updates was a bit difficult to follow at first. Some of the notation such as using $\Delta_{Exp}$ was slightly confusing since it suggests that it is the overall expected update even though it's only the expected update over $\xi_a$. An alternative could be $\Delta_{\mu_a}$. The expectation $E_{s,a$}[\Delta \theta(s,a)]$ is slightly unclear as to what distribution is used for $(s,a)$ here.

**Questions:**

- Why does aligning the gradients w.r.t the action of the Q-value and the TD error make sense? Is there an interpretation for this? It seems like the inner product between those gradients could be a meaningful quantity more broadly e.g. as an evaluation metric for the quality of a critic.
- What is $\eta$ in equation (11)? It wasn't described in the text or the appendix.
- This method could potentially also be used in cases when the simulator is differentiable to start with. Have you experimented with this at all?
- Minor point: I would consider modifying the title of the paper to a more descriptive name which could reference model-based RL or Dyna-style updates.


**Limitations:**

The limitations are discussed.

---

> ### Author Rebuttal · Authors · 2023-08-09
>
> We thank the reviewer for the helpful comments!
>
> > For example, I would avoid using the term "Monte-Carlo" estimates when discussing sampling of TD updates since it can lead to some confusion due to "Monte-Carlo" often being used to refer to MC estimates of the return in contrast to the bootstrap estimates that TD uses. I would consider using the phrase "sampled TD update" instead of "MC TD update". E.g. line 31, line 42, ...
>
> Thanks for spotting this clash with pre-existing RL terminology.  We will switch to your suggested terminology.
>
>
> > I would also suggest including a line about continuous state and actions in the introduction to clarify the problem setting since the proposed algorithm would mainly be applicable there.
>
> Thank you for the suggestion. This is included in the original abstract, and we will include a note about it in the Introduction too.
>
>
> > Section 3.1 describing the updates was a bit difficult to follow at first. Some of the notation such as using $\Delta_\text{Exp}$ was slightly confusing since it suggests that it is the overall expected update even though it's only the expected update over $\Xi_\text{a}$. An alternative could be $\Delta_{\mu_a}$.
>
> We'll consider these notational changes.  We hoped that the $(s, \mu_\text{a})$ function arguments in $\Delta_\text{Exp}(s, \mu_\text{a})$ would emphasise that this is taking an expectation only over the noise in the action, for fixed state, $s$ and mean action, $\mu_\text{a}$.
> The issue with $\Delta_{\mu_a}$ is that we contrast the exact expectation, $\Delta_\text{Exp}$, with the Taylor-series approximation, $\Delta_\text{Ta}$. But they're both exact/approximate expected updates over $\Xi_\text{a}$ with fixed $\mu_a$, so $\Delta_{\mu_a}$ wouldn't make the required distinction.
>
> We could change $\Delta_\text{Exp} \rightarrow \Delta_\text{Exact}$?  Alternatively, we could further clarify the definition of $\Delta_\text{Exp}$ in Eq. 7?  Let us know if you have a preferred option.
>
>
>
> > The expectation $E_{s,a}[\Delta \theta(s,a)]$  is slightly unclear as to what distribution is used for here.
>
> We will add a sentence clarifying the distribution here. To clarify, this distribution refers to the overall visited state distribution, $s \sim d^\pi$, and the initial policy distribution, $a \sim \pi^\text{init}$.
>
>
>
>
> > Why does aligning the gradients w.r.t the action of the Q-value and the TD error make sense? Is there an interpretation for this? It seems like the inner product between those gradients could be a meaningful quantity more broadly e.g. as an evaluation metric for the quality of a critic.
>
> First, it is important to emphasise that our updates should just be understood as the expectation of standard TD updates (Eq. 7) over a distribution over initial actions, $a$, centred at $\mu_\text{a}$. In that context, it is possible to understand how the "alignment" emerges.  Specifically, lets consider a random action in the direction of $\nabla_a \delta_\theta$, e.g. $a = \mu_\text{a} + \epsilon \nabla_a \delta_\theta$.  That action has a bigger, positive prediction error (as we've moved along the gradient of $\delta_\theta$). And under standard TD updates, a bigger positive prediction error should lead to a bigger positive update to $Q_\theta$.  This interaction of bigger positive prediction errors leading to bigger positive updates to $Q_\theta$ implies an "alignment" intuition.
>
> Its a super-interesting idea to use this quantity as an evaluation metric!  It certainly could work!  But we worry that doing a sufficiently thorough investigation of the properties of this metric is a significant piece of work requiring its own paper.
>
>
> > What is $\eta$ in equation (11)? It wasn't described in the text or the appendix.
>
> Good catch!  We have deleted the $\eta$'s.  (They were intended to represent a learning rate in a previous version of the derivation, but we subsequently removed it to simplify the derivations).
>
>
> > This method could potentially also be used in cases when the simulator is differentiable to start with. Have you experimented with this at all?
>
> Agreed, Taylor TD has broad applications to differentiable simulator settings. For instance, [1] highlights the importance of learning accurate value functions with differentiable simulator in order to perform policy updates beyond a short horizon of returns. Taylor TD would greatly aid the process of learning accurate value functions with little-to-none added computational cost to the differentiable simulator. However, in the current manuscript, we decided to focus on the usual (and potentially harder) RL setting where the model must be learned from the transitions (e.g. as in Janner et al., 2019 - MBPO).
> We'd be happy to run any additional experiments if the reviewer can suggest a specific setting.
>
> [1] Xu, J., Makoviychuk, V., Narang, Y., Ramos, F., Matusik, W., Garg, A., \& Macklin, M. (2022). Accelerated policy learning with parallel differentiable simulation. arXiv preprint arXiv:2204.07137.
>
>
> > Minor point: I would consider modifying the title of the paper to a more descriptive name which could reference model-based RL or Dyna-style updates.
>
> Something like "Taylor TD-learning in model-based reinforcement learning"?  We'd be happy to make that change.

---

> > ### Comment · Reviewer_9ew5 · 2023-08-18
> > **Response**
> >
> > Thank you for the response and consideration.
> >
> > About the presentation:
> > > Alternatively, we could further clarify the definition of  in Eq. 7? Let us know if you have a preferred option.
> > Perhaps clarifying the definition would be sufficient and re-iterating that $\mu_a$ is fixed and the only randomness is over $\Xi_a$. I agree there's some difficulty in choosing the right notation here and leave it to your discretion.
> >
> > >Something like "Taylor TD-learning in model-based reinforcement learning"? We'd be happy to make that change.
> > That sounds good.
> >
> > I appreciate the explanation of the alignment of TD-error and $Q$ value gradients. It makes sense to me although it doesn't necesssarily explain why this direction would be useful for policy optimization. There may be some deeper reason here to be found.
> >
> > I am satisfied with the response and the references to risk-sensitive RL by Reviewer 9c2H would be a nice addition too. I would also encourage some investigation regarding the alignment of TD-error and $Q$ values to be done even if it is preliminary.
> > Overall, I will still recommend acceptance.

---

### Official Review · Reviewer_wJvf · 2023-07-24

**Soundness:** 2 fair
**Presentation:** 2 fair
**Contribution:** 2 fair
**Rating:** 6
**Confidence:** 3

**Summary:**

This paper,
1. Proposes Taylor TD, a model-based RL algorithm that uses a Taylor series expansion to analytically estimate the expected TD update over a distribution of nearby state-action pairs. This reduces variance compared to standard MC TD updates.
2. Provides theoretical analysis showing the variance of Taylor TD updates is lower than standard TD updates. Also proves stability of Taylor TD with linear function approximation.
3. Empirically demonstrates lower variance of Taylor TD on several RL tasks.
4. Combines Taylor TD with TD3 in an algorithm called TaTD3. Shows strong performance of TaTD3 relative to model-free and model-based baselines on MuJoCo benchmarks.


**Strengths:**

1. The core idea of using Taylor expansions to estimate expected TD updates is novel and well-motivated from a theoretical perspective. This analytic approach to reducing variance is interesting.
2. The method preserves the convergence guarantees of TD-learning under linear function approximation, as shown formally. This is an important theoretical contribution.
3. The variance reduction analysis provides evidence Taylor TD reduces variance versus standard TD methods.
4. The algorithm is straightforward to implement on top of existing model-based RL frameworks.


**Weaknesses:**

1. In Figure 2, Taylor TD3 seems to provide only very small performance improvements over the baselines in most environments (4 out of 6). The gains are noticeable in only 2 environments. Given that Taylor TD3 is much more complex, is the minimal improvement in the majority of environments concerning? The results seem to imply the benefits may be limited to certain environments. Some discussion of why the gains are so marginal in certain tasks would be useful.
2. Some derivations are not clear. It is not clear what loss function are the authors using to learn the model.


**Questions:**

1. In the Taylor TD update derivations (Equations 12, 17 to 18), the Hessian terms present after the Taylor expansions disappear in the final update equation. Can you clarify the assumptions or steps that lead to the Hessian terms dropping out? As these terms arise directly from the Taylor approximations but then vanish, an explicit explanation should be provided about how and why they drop out of the final update form.
2. It is also not clear how the loss function (equation 18) is derived. Could you please provide the derivation?
Algorithm 1 includes an update equation for the model parameters w using some loss L, but does not define what this loss function L is (unless I am mistaken). Please specify what objective or loss function is used to optimize the model parameters w based on the observed environment transitions.
3. The current evaluations of Taylor TD are limited to a small set of MuJoCo continuous control tasks. Did the authors consider testing on additional environments that evaluate aspects like partial observability, sparse rewards, and higher task dimensionality? A more diverse test suite could provide greater insight into the strengths and weaknesses of Taylor TD across different conditions. What other test beds do you think could be useful for thorough analysis? What are the challenges that you foresee?
4. The hyperparameters indicate high values for \lambda_a and very low values for \lambda_s. Why is that the case?


**Limitations:**

The authors mentioned about the limitations in the last section and I do not forsee any negative societal impacts.

---

> ### Author Rebuttal · Authors · 2023-08-09
>
> We thank the reviewer for the helpful comments!
>
> > In Figure 2, Taylor TD3 seems to provide only very small performance improvements
>
> While we agree with the reviewer's assessment, it is important to note that there is no single method that performs competitively with TaTD3.
> Particularly, MBPO is competitive with TaTD3 on HalfCheetah-v2, Walker2d-v2 and Ant-v2, but learns extensively slower on the most complex environment, Humanoid-v2.
> In contrast, MAGE performs well on HalfCheetah-v2 and the most complex environment, Humanoid-v2, but fails in other environments: Walker2d-v2, Ant-v2 and Hopper-v2.
> In contrast, TaTD3 performs consistently (close to) the best in all these environments.
> It should also be noted that Taylor TD3 is not necessarily more complex than the two strongest baseline algorithms, MBPO and MAGE.
> For instance, although MBPO does not require any extra gradient term, it uses additional compute to iterate through the model predictions over multiple steps for each update (up to 25x).
>
>
> > Some derivations are not clear. It is not clear what loss function are the authors using to learn the model.
>
> We thank the reviewer for pointing this out. We neglected to include a section in the appendix where we described how the model was learned. To clarify, the reward and transition model were learned by maximum likelihood based on the observed transition, as for most model-based RL approaches (e.g. Janner et al., 2019 - MBPO). A section describing this process will be added in the appendix of the camera-ready version of the paper.
>
>
>
> > In the Taylor TD update derivations (Equations 12, 17 to 18), the Hessian terms present after the Taylor expansions disappear
>
> We have a choice about the order of the Taylor expansion.  For instance, we could choose to do a first-order expansion, which would include gradient but not Hessian terms, or we could choose to do a second-order Taylor expansion, which would include Hessian terms.  We choose the first-order approach.  We did play with the second-order terms, but we found they added considerable additional complexity for little obvious benefit.  Additionally, we were able to get theoretical guarantees on the first-order approach (Appendix B), and it wasn't clear we'd be able to get the same guarantees on the more complex second-order methods.
>
> By the Hessian, are you referring to $\Sigma_\text{a}$, which is present e.g. in Eq 11, but not in Eq 12, 17 to 18?  $\Sigma_\text{a}$ isn't the Hessian, it is the covariance of the Gaussian distribution over initial actions, $a$, in the Bellman update (see Eqs 4-6).  We get to choose this covariance, and in going from 11 to 12, we choose $\Sigma_\text{a} = \lambda_\text{a} I$. We will re-emphasise this point in the camera-ready text.
>
>
> > It is also not clear how the loss function (equation 18) is derived.
>
> The key derivation of the critic updates is in Sec. 3.1 (Eq. 12) for the actions and Sec. 3.2 (Eq. 17) for the states (see Appendix A for more details).  To get to Eq. 18 requires two steps.
> First, we apply the usual RL implementation trick, of writing the updates in terms of a gradient of a loss, with stop-gradients operations to avoid taking the gradient of $\delta$ (Appendix D).
> Second, we replace the dot product in Eq 12 with a cosine similarity.
> This gives a useful normalization, which does seem to improve performance (see Appendix I.2).
> We agree that these steps were not described clearly enough in the main text, so we will update the camera-ready to spell this out.
>
> The reward and transition model were learned by maximum likelihood based on the observed transition, as in most model-based RL approaches (e.g. as in Janner et al., 2019 - MBPO). We will add an Appendix describing the model-learning to the camera-ready.
>
>
>
> > The current evaluations of Taylor TD are limited to a small set of MuJoCo continuous control tasks.
>
> We agree with the reviewer that a more diverse test suite could provide greater insight into the strengths and weaknesses of Taylor TD across different conditions.
> Partial observability is complex, as it requires latent states in the model.  We suspect it should be possible, but anticipate that it is a "research-project" level exercise that is out of scope for this paper.
> Additionally, Humanoid is typically considered a high dimensional control task with the RL literature, given the 376 dimensional state space and the 17 dimensional action space.
> It should also be noted the set of MuJoCo continuous control tasks that we chose represent the standard benchmark on which RL algorithms are tested for continous control. For instance, the baseline algorithms we tested Taylor TD against were themselves tested on 6 similar MuJoCo continous control environments in the original papers and this is the case for most continuous control RL algorithms (e.g., SAC, TD3).
> Nevertheless, we are happy to run further experiments for camera-ready if you can suggest specific benchmarks.
>
>
>
> > The hyperparameters indicate high values for \lambda_a and very low values for \lambda_s.
>
> In general, we found the action-based expansion to bring the largest benefits to performance, although the state-based expansion is still useful (i.e., see Appendix I.1). We think one reason for this may be because, we know that Gaussian distributions over $a$ work well when evaluating the Bellman updates.  However, we don't know whether the same applies for the distribution of state; a Gaussian distribution may only work well for very nearby states.
> We should also stress that any direct comparison between the magnitude of the state covariance (\lambda_s) and the action covariance (\lamba_a) may not be very meaningful.
> This is because the scale of the actions and states tend be fairly different in the tested environments, making any comparison between $\lambda_s$ and $\lambda_a$ harder to assess. We will add this key discussion in section A.6. of the camera-ready.

---

> > ### Comment · Reviewer_wJvf · 2023-08-17
> > **Thank you**
> >
> > Dear Authors,
> >
> > I appreciate the thorough responses to all my questions. My concerns on derivation, loss function and others have been addressed. I adjusted the score accordingly.
> >
> > Thanks

---

### Official Review · Reviewer_9c2H · 2023-07-29

**Soundness:** 2 fair
**Presentation:** 2 fair
**Contribution:** 3 good
**Rating:** 5
**Confidence:** 3

**Summary:**

The author presents a variance reduction trick in TD learning by taking the analytical expectation of the gradient using first order taylor expansion. Standard RL, replaces the expectation over the gradient of critic (Q-value) with a sampled value from the replay buffer or online learning samples. The paper proposed to replace this expected gradient of Q value function with first order taylor series expansion of the same. They proposed taylor expansion over both state and action.


** I have read the rebuttal of the authors.

**Strengths:**

Originality
1. The work is proposing novel research in the direction of stabilizing the critic updates by reducing the variance using first order taylor approximation of the expected gradients.

Quality
1. The paper could be improved on how the experiments are conducted for Fig 1. Furthermore provide better clarity on the questions section later.

Significance
1. The work is definitely targeting a significant problem that could have impact on RL algorithms in general. This work is stabilizing the critic updates by reducing the variance.

**Weaknesses:**

The weakness/questions are mentioned in the next section. If the authors provide justification for the below, I am willing to change the score.

Add references for the other variance reduction work in direction of - variance reduction of policy updates, inherent stochasticity (risk-sensitive RL). They are very related to your proposed research.

**Questions:**

1. It is not clear in the paper how the model (transition, reward) is learnt using the $L_\theta$ objective in Eq 18?  Algo 1 shows the update $w$ (model parameter ) with the gradient of $L_\theta$ without showing how $w$ influences it.
2. Why is $\mu_a$ in Eq 19 sampled from deterministic target policy? Why do you enforce this determinism on the policy being learnt?
3. In Fig 1, what is the variance computer over? Is it same $(s,a)$ pair and then compute multiple gradients over that? Is it batch data, comprising of same data for both MC TD and Taylor-TD and then the variance over updates is computed? Because if the variance is computed over different state-action, then the high variance could be a factor of that (s,a) pair having high stochasticity in the dynamics. It wouldn’t mean necessarily that updates have less variance because of taylor approximation. Please explain what this variance update is over.
4. What is the effect of reducing the variance in critic update vs to that actor update(Ref[1])? Have you tried ablation study for the same - which one is better, or what effects they have on learning separately vs together?
5. Provide some connections with Risk-sensitive RL and robust RL - inherent stochasticity vs imperfect knowledge? Will be good to connect to the literature in that area (add references for same)!
6. Comparison with expected sarsa kind of update in tabular environments? That would help to understand how expected update over the gradients help over Expected sarsa. (Would be good to know more insights on this, just something additional)

References --

[1] Variance Reduction for Policy-Gradient Methods via Empirical Variance Minimization

**Limitations:**

Yes the authors have included the limitation section. The negative societal impact section doesn't apply to this line of research.

---

> ### Author Rebuttal · Authors · 2023-08-09
>
> We thank the reviewer for the helpful comments!
>
> > Add references for the other variance reduction work ...  Risk-sensitive RL and robust RL
>
>
> Thanks for these suggestions! We will add your reference [1], and we did a literature review on risk-sensitive RL approaches (e.g., [2],[3],[4],[5]), which we will add to the camera ready. Please do suggest other papers!
>
> We agree Taylor TD may be applicable to risk-sensitive RL. In the presence of a good model of the transitions, Taylor TD may can approximate the values of risky actions around safe (e.g. deterministic) actions, without actually needing to take those actions, thanks to the Taylor expansion of the TD objective.
> That said, Taylor TD does not directly estimate the uncertainty induced by imperfect knowledge (i.e., epistemic uncertainty), so it may be less applicable to robust RL.
> We would also like to stress that in the related work section of the original manuscript, we discussed Expected Policy Gradients (Ciosek and Whiteson, 2018), Mean Actor Critic (Asadi et al., 2017) and "all-action" policy gradient (Petit et al., 2019) as related methods. These methods tackle the variance at the level of the policy rather than critic updates by also integrating over the stochasticity induced by the action distribution.
>
> [1] Variance Reduction for Policy-Gradient Methods via Empirical Variance Minimization
>
> [2] Tamar et al. Policy gradients with variance related risk criteria. ICML (2012)
>
> [3] Bellemare et al. A distributional perspective on reinforcement learning. ICML (2017)
>
> [4] Lim & Malik. Distributional Reinforcement Learning for Risk-Sensitive Policies. NeurIPS (2022)
>
> [5] Fu.  Risk-Sensitive Reinforcement Learning via Policy Gradient Search. arXiv (2018).
>
> > It is not clear in the paper how the model (transition, reward) is learnt
>
> We thank the reviewer for pointing this out. We neglected to include a section in the appendix where we described how the model was learned. To clarify, the reward and transition model were learned by maximum likelihood based on the observed transition, as in most model-based RL approaches (e.g., as in Janner et al., 2019 - MBPO). We will add an Appendix describing the model-learning to the camera-ready. Note, Eq 18 refers to the critic loss (derived from the Taylor state and action expansion).
>
> > Why is $\mu_\text{a}$ in Eq 19 sampled from deterministic target policy?
>
> There are really two separate questions here.
>
> First, there is the question of whether $a$, the initial action in the TD update, is stochastic/deterministic. In our setting, $a$ is always stochastic, which helps learn a broader Q function with performance benefits over deterministic initial actions (e.g., see the poor performance of Dyna-TD3 baseline algorithm).
> However, stochastic initial actions also increase the TD-update variance relative to deterministic initial actions.
> Taylor-TD mitigates this increased variance by analytically approximating the expected TD update arising from stochastic initial actions.
>
> Second, there is the question of whether the target policy (which we use to generate $a'$ in the TD update) is stochastic/deterministic.  It turns out the Taylor TD can be used with deterministic or stochastic target policies. We used deterministic target policies, because in a deterministic environment, the optimal policy is deterministic (or at least, _an_ optimal policy is deterministic).
>
> > In Fig 1, what is the variance computer over?
>
> We agree that this wasn't quite clear enough in the original manuscript, and will clarify in the camera-ready. We used exactly the same initial state-action pairs when comparing standard (MC) TD and Taylor-TD updates. The additional variance in standard (MC) TD updates comes from the usual approach of adding a small amount of Gaussian noise to the initial action, $a$, in the TD update; while in Taylor-TD, we analytically integrate over a Gaussian distribution over actions (and states).
>
> > What is the effect of reducing the variance in critic update vs to that actor update(Ref[1])?
>
> We thank the reviewer for bringing this paper to our attention, we will add it to the related work.
> We would expect that the improvements from each method would compound, as one is reducing variance in the actor update, while the other is reducing variance in the critic update: we will attempt to perform this experiment for the camera-ready deadline.
>
> That said it is important to note an additional complication.  In particular, TaTD can be used with deterministic or stochastic target policies, and our current experiments use deterministic target policies, as we are mostly working in deterministic environments.  However, [1] can offer no benefits with deterministic target policies, as [1] reduces variance from stochastic target policies.
>
> > Comparison with expected sarsa kind of update in tabular environments?
>
> First, as noted in the Abstract, Taylor-TD requires continuous state and action spaces, for the required derivatives, so the connection to tabular settings is unclear.
>
> More fundamentally, expected SARSA computes the expected $Q(s', a')$ under the distribution over _next_ actions, $a'$, given the current policy.
> In contrast, Taylor-TD computes the expected TD update under a distribution over _initial_ action, $a$.
> Thus, it is possible to combine expected SARSA and Taylor-TD, and as they reduce different components of the variance, we would expect them to be complementary.
> However, it is important to note that expected SARSA only makes sense when we have a stochastic policy, which implies a need to compute the expectation of $Q(s', a')$ under a distribution over _next_ actions, $a'$.
> In contrast, Taylor-TD allows the use of either stochastic or deterministic.
> As noted above, we use deterministic target policies (as an optimal policy in a deterministic environment is deterministic). And if we use a deterministic target policy, there is no need for an expectation over next actions: we can just compute $Q(s', a')$.

---

> > ### Comment · Reviewer_9c2H · 2023-08-17
> > **Thanks for the responses**
> >
> > Thanks for addressing some of my concerns. I had some additional input. Please clarify more on the third point mentioned below.
> >
> >
> > 1. "These methods tackle the variance at the level of the policy rather than critic updates by also integrating over the stochasticity induced by the action distribution."
> >
> > -> There exists work in risk-sensitive RL literature that tackles variance estimation by directly using the Bellman operator for the variance. Would be good to include them in references too. Also it would be good to provide clarity on how it differs from these works. I agree that your work minimizes the variance of the value function estimation by using Taylor approximation.
> >
> > [1]Sobel, M. J. 1982. The variance of discounted Markov decision processes. Journal of Applied Probability 19(4): 794– 802.
> >
> > [2]Jain, Arushi, et al. "Variance penalized on-policy and off-policy actor-critic." Proceedings of the AAAI Conference on Artificial Intelligence. Vol. 35. No. 9. 2021.
> >
> > [3]Tamar, A.; Di Castro, D.; and Mannor, S. 2013. Temporal difference methods for the variance of the reward to go. In International Conference on Machine Learning, 495–503.
> >
> > 2. “It is not clear in the paper how the model (transition, reward) is learnt”
> >
> > -> If the transition and reward are learnt using maximum likelihood, please change the notation in Algo 1, the model update step where the same \mathcal{L} was used. The same \mathcal{L} is used in Eq 18 which makes it very confusing. Further, also describe the notation in intro/Taylor TD section - how the model was learnt.
> >
> > 3. Why is the standard TD method called (MC) TD? Because in the TD method, we are still estimating the target by bootstrapping with a 1-step Q value. I can't understand why MC is there. In this work, the “standard TD style” is also used. Could you clarify the difference between the two?

---

> > > ### Author Response · Authors · 2023-08-19
> > > **Response**
> > >
> > > Thanks for your continued engagement!
> > >
> > > [1,2,3] look like great papers.  In fact, they're super-relevant for some of our other work, so I will pass them on.  And of course, we will include them in the related work for this paper.  But importantly they are doing something quite different from this paper.  [1,2,3] are all considering variance _in the return_, induced either by stochastic _rewards_ or _transitions_.  In contrast, this paper is not considering variance in the return, nor does it considering variance that arises through stochastic rewards or transitions.  Instead, we are considering variance in the _TD update_ induced by stochasticity in the _choice of initial action_ (and visited states).
> > >
> > > [1] Sobel, M. J. 1982. The variance of discounted Markov decision processes. Journal of Applied Probability 19(4): 794– 802.
> > >
> > > This paper presents formulas "for the variance and higher moments of the present value of single-stage rewards in a finite Markov decision process".
> > >
> > > [2] Jain, Arushi, et al. "Variance penalized on-policy and off-policy actor-critic." Proceedings of the AAAI Conference on Artificial Intelligence. Vol. 35. No. 9. 2021.
> > >
> > > This paper proposes "on-policy and off-policy actor-critic algorithms that optimize a performance criterion involving both mean and variance in the return."
> > >
> > > [3] Tamar, A.; Di Castro, D.; and Mannor, S. 2013. Temporal difference methods for the variance of the reward to go. In International Conference on Machine Learning, 495–503.
> > >
> > > (The title is a reasonable summary.)
> > >
> > >
> > > > If the transition and reward are learnt using maximum likelihood, please change the notation in Algo 1, the model update step where the same \mathcal{L} was used. The same \mathcal{L} is used in Eq 18 which makes it very confusing. Further, also describe the notation in intro/Taylor TD section - how the model was learnt.
> > >
> > > Good catch.  We'll definitely update this as part of the more general revisions regarding the model learning. In particular, we will use $\mathcal{L}^{\text{model}}$ as the model loss, and $\mathcal{L}^{\text{critic}}$ as the critic loss (e.g. in Eq. 18).
> > >
> > > > Why is the standard TD method called (MC) TD? Because in the TD method, we are still estimating the target by bootstrapping with a 1-step Q value. I can't understand why MC is there. In this work, the “standard TD style” is also used. Could you clarify the difference between the two?
> > >
> > > We're using Monte-Carlo (MC) as opposed to Taylor (Ta) to emphasise that there are two ways to compute the expectation in Eq. 7.  We could draw many samples of $\xi_i$ and compute an empirical average.  That's a Monte Carlo approach.  Alternatively, we could use our proposed approach of computing the analytic expectation for a first order Taylor series expansion.
> > >
> > > We realise that there is a bit of a clash with an alternative use of "Monte-Carlo" in RL (specifically Monte-Carlo estimates of the return).  Following Reviewer 9ew5's suggestion we will switch to using "sampled TD update" rather than "MC TD" to avoid this potential source of confusion.

---

> > > > ### Comment · Reviewer_9c2H · 2023-08-21
> > > > **Reviewer response**
> > > >
> > > > Thank you for your clarifications regarding the notations and experiments. I've revised my score now. Your paper presents a substantial technical contribution to the field. However, refining the writing will greatly enhance its accessibility to a wider audience. As a fellow reviewer, I also needed some time to fully understand the core contributions. I strongly encourage the authors to focus on improving the readability, as this will undoubtedly solidify the paper's strength. Your work is really good, and with some writing adjustments, it can become amazing.

---

### Official Review · Reviewer_cy77 · 2023-07-30

**Soundness:** 3 good
**Presentation:** 3 good
**Contribution:** 3 good
**Rating:** 6
**Confidence:** 2

**Summary:**

The authors are proposing a method for reducing the variance of TD-learning updates for model-based RL applied to problems with continual state-action spaces. The method relies on Taylor expansion of noise terms in action and initial state distribution, reducing the contribution of those to the variance of TD-update. The authors demonstrate variance reduction both theoretically and empirically, as well as performance on par with SOTA model-based methods such as MBPO (Janner et al, 2019) and MAGE (D'Oro et al, 2020). For empirical evaluation, the authors apply the proposed update adjustment to TD3 with Dyna.

**Strengths:**

The empirical results seem pretty strong, though I would have found results with longer training times to be more convincing (about 2x longer, i.e. as long as was used in other work such as Janner et al, 2019).

I appreciated the strong theoretical backing, i.e. demonstrating lower variance and stability guarantees (though it should be noted that I did not check the proofs as this work falls outside of my expertise).



**Weaknesses:**

Minor:

- the authors should be clearer in the main body of the paper about the additional computational demands of the proposed method

- some of the particularly interesting results can be only found in the Appendix. For example, the claim that the method performs better on large state-action spaces seems like an important one, and it was good to see it explored in a more controlled setting in Appendix F. Similarly, we can only find ablations (importance of state expansion and cosine similarity) in Appendix I (also it would be good to see these results on all 6 environments). I hope the authors can find a way to move these results to the main body of the paper (personally I found those to be more insightful than the comparison in Fig 3).

**Questions:**

1. In Figure 1, does the variance of each method change during training?

2. In Figure 2, how do the results look like if the models are trained for another 2x steps? What about comparison in terms of training time?



**Limitations:**

No concerns regarding potential negative societal impact, and the limitations were discussed to a reasonable amount.

---

> ### Author Rebuttal · Authors · 2023-08-09
>
> We thank the reviewer for the helpful comments!
>
> > the authors should be clearer in the main body of the paper about the additional computational demands of the proposed method.
>
> We raised this point in the Limitations section of the original manuscript, and provided a reference to the Appendix for the actual computational costs in terms of training time, highlighting the additional computational cost is not that large (i.e., standard TD learning is only 20\% faster on average). We will bring this table to the main text in the camera-ready.  However, standard TD learning is perhaps the fastest baseline.  Additional analysis showed that while Taylor-TD was a bit slower than TD, it was much closer to MAGE (for Walker 2d, Taylor-TD took 68 s while MAGE took 63 s).
>
> > In Figure 2, how do the results look like if the models are trained for another 2x steps? What about comparison in terms of training time?
>
> These runs require quite a bit of additional compute time, so we have prioritised Ant, as that's the only one that seems not to have saturated yet. In line with the rest of the results for Ant, we found that MBPO (5260 at 250k steps) seemed to be doing a bit better than TaTD3 (4860 at 250k steps).  For the camera-ready, we will run Ant out to 300k steps, and increase the number of iterations in the rest of the environments.
>
> > some of the particularly interesting results can be only found in the Appendix. For example, the claim that the method performs better on large state-action spaces seems like an important one, and it was good to see it explored in a more controlled setting in Appendix F. Similarly, we can only find ablations (importance of state expansion and cosine similarity) in Appendix I (also it would be good to see these results on all 6 environments). I hope the authors can find a way to move these results to the main body of the paper (personally I found those to be more insightful than the comparison in Fig 3).
>
> Thanks!  We agree that the Appendix contains a number of interesting results, and to the extent that is possible within space constraints, we will move some of these results into the main text.
>
>
> > In Figure 1, does the variance of each method change during training?
>
> We have run preliminary experiments assessing the variance reduction from Taylor TD at different stages of training.  We found that for completely untrained networks, there was little benefit, likely because the untrained model provides poor gradient estimates.  However, a beneficial variance reduction, similar in magnitude to those in the main-text, emerges early on in training, and remains for the rest of the training run.  For the camera-ready paper, we will modify Fig. 1 to show the full time-course of the variance reduction through training.

---

> > ### Author Response · Authors · 2023-08-19
> > **More details about runtimes**
> >
> > We have got some numbers around the runtimes for our method (TaTD3) against the competitive baselines (MBPO and MAGE).  MAGE has about the same runtime as our method (TaTD3), while MBPO is a lot slower.
> >
> > | Method | Pendulum | Walker | Ant   | Humanoid |
> > | --- | ---- | ---- | --- | --- |
> > | MAGE   | 36 s     | 63 s   | 75 s  | 127 s    |
> > | MBPO   | 52 s     | 133 s  | 158 s | 235 s    |
> > | TaTD3  | 38 s     | 68 s   | 72 s  | 117 s    |

---

### Comment · Area_Chair_BAJo · 2023-08-18
**Completing paper discussion**

Dear Reviewers,

Thank you for your efforts here. If you haven't already, please acknowledge the authors responses. Also please read other reviews and let the authors know if additional concerns have been raised.

Reviewer 9c2H: Thank you for engaging in discussion to see if the authors can clarify your concerns!

Thanks,
Your AC

---

### Decision · Program_Chairs · 2023-09-21

**Decision:**

Accept (poster)

**Comment:**

This paper studies the problem of variance in the temporal difference (TD) update that is widely used in RL. The paper proposes a method for reducing variance based upon an analysis using the Taylor expansion of the TD update. The method requires the use of a learned state transition model and hence can be seen as a hybrid model-free / model-based RL method. The paper primarily shows that the method reduces variance and leads to more data efficient RL through empirical analysis in standard continuous control RL tasks.

Main Strengths: the method is novel and the evaluation of it is sound and shown to provide empirical benefit for an important operator in RL. The proposed method is motivated from solid theoretical analysis.

Main Concerns: Main concerns from the reviewers were lack of clarity in some places and derivations. These can be fixed in the camera ready revision. One reviewer pointed out a number of references that the authors should decide whether they want to discuss or not in the revised version. I would add "Reducing Sampling Error in Batch Temporal Difference Learning" [Pavse et al. 2020] as another paper that seems related (though not conflicting with this paper's novelty) and the authors should review it and consider its relevance. Missing citations and need for some clarifications were not major concerns.

Overall, this paper presents a theoretially motivated method for improving the TD update that leads to empirical improvement in deep RL experiments. Given the importance of the TD update in many RL algorithms I believe this paper makes a valuable contribution to the community.